# Periodontitis induces skeletal muscle atrophy by increasing circulating levels of activin A.

Wonn Shim [1], Joonho Suh [1], Ha Kyeong Kim [2,3], Na-Kyung Kim[1], Yongbaek Kim [4,5,6], Yong Taek Jeong[2,3], Min-Sung Kim[7], Se-Jin Lee[8,9] & Yun-Sil Lee [1] ✉

Periodontitis is linked to various systemic conditions, but its impact on skeletal muscle remains unclear. Here, we utilized a ligature-induced periodontitis model in male mice and showed that periodontitis significantly reduces muscle and bone mass without affecting fat mass or food intake. Interestingly, activin A, well-documented inducer of muscle atrophy, is highly expressed in periodontitis-affected gingiva. The activin A gene (*Inhba*) is predominantly expressed in gingival fibroblasts and epithelial cells, which undergo significant proliferation as periodontitis progresses, as well as in myeloid cells infiltrating inflamed periodontal tissues and myeloid cell-derived osteoclasts. A similar upregulation pattern of *INHBA* was also confirmed in periodontitis-affected human tissues by scRNA-seq analysis. Furthermore, we demonstrated that serum activin A levels are increased in periodontitis-affected mice and patients. Gingival overexpression of activin A via AAV-*Inhba* transduction activates canonical activin signaling in skeletal muscle, as evidenced by increased pSMAD3 and MuRF1 expression, leading to significant muscle loss. Notably, intra-gingival injection of si*Inhba* significantly reduced serum activin A levels and restored muscle mass and myofiber size. Our findings indicate that activin A is a mediator of muscle atrophy in periodontitis and suggest that local injection of si*Inhba* may prevent periodontitis-induced muscle atrophy without apparent systemic adverse effects.

Periodontitis is a common disease accompanied by the destruction of periodontal tissues due to excessive inflammation. Nearly half of adults over the age of 30 are affected, making periodontitis a significant public health concern. While research has mostly focused on its localized effects, such as inflammation, oral dysbiosis, and tissue destruction, growing evidence suggests that these local effects lead to systemic consequences. For instance, dissemination of periodontal bacteria into the bloodstream results in bacteremia, and sustained inflammation contributes to low-grade systemic inflammation, potentially impacting cell fate in the bone marrow[1]. Furthermore,

[1]Department of Molecular Genetics, School of Dentistry and Dental Research Institute, Seoul National University, Seoul, South Korea. [2]BK21 Graduate Program, Department of Biomedical Sciences, Korea University College of Medicine, Seoul, South Korea. [3]Department of Pharmacology, Korea University College of Medicine, Seoul, South Korea. [4]Research Institute for Veterinary Science, College of Veterinary Medicine, Seoul National University, Seoul, South Korea. [5]BK21 FOUR Program for Future Veterinary Medicine Leading Education and Research Center, College of Veterinary Medicine, Seoul National University, Seoul, South Korea. [6]Laboratory of Clinical Pathology, College of Veterinary Medicine, Seoul National University, Seoul, South Korea. [7]Department of Life Sciences, Pohang University of Science and Technology, Pohang, South Korea. [8]The Jackson Laboratory for Genomic Medicine, Farmington, CT, USA. [9]Department of Genetics and Genome Sciences, University of Connecticut School of Medicine, Farmington, CT, USA. ✉e-mail: yunlee@snu.ac.kr

periodontitis has been associated with numerous systemic conditions, including cardiovascular disease, diabetes, Alzheimer's disease, and adverse pregnancy outcomes[2].

More recently, the effects of periodontitis on the musculoskeletal system have been investigated, primarily focusing on its impact on the bone. For example, the association between periodontitis and osteoporosis has been explored[3], and studies using experimental periodontitis models in mice have demonstrated that periodontitis can induce systemic bone loss[4]. Additionally, pathogens responsible for periodontitis have been reported to release extracellular vesicles that promote alveolar bone resorption, further contributing to systemic bone loss[5]. In contrast, the effects of periodontitis on muscle health are still poorly understood. Because maintaining muscle mass during systemic bone loss may increase susceptibility to fractures, certain molecules released during bone resorption could potentially contribute to muscle weakening. However, no specific molecules related to this phenomenon have been reported. Some studies suggest that the oral administration of periodontal pathogens or their derived lipopolysaccharides (LPS) may negatively impact muscle health[6,7], but the experimental conditions in these studies differ significantly from typical periodontitis, leaving its effects on muscle mostly unresolved.

Sarcopenia is a debilitating, age-related condition characterized by excessive muscle loss and weakness, which increases the risks of frailty, falls, and mortality. With no FDA-approved treatments available, it is crucial to prevent sarcopenia by identifying its contributing factors. Chronic inflammatory diseases, such as chronic kidney disease (CKD) and chronic obstructive pulmonary disease (COPD), are recognized contributors to muscle atrophy, suggesting that periodontitis, another chronic inflammatory condition, could similarly contribute to muscle deterioration. A prospective cohort study by Hämäläinen et al. supports this link, showing that older adults with periodontitis experienced reduced handgrip strength over five years[8]. While this study provides initial evidence, further studies are needed to understand the underlying mechanisms, which could inform new strategies for sarcopenia management.

In this study, we utilized an experimental periodontitis model in male mice to investigate the direct relationship between periodontitis and the musculoskeletal system, demonstrating that periodontitis induces systemic muscle and bone loss. Notably, we found that activin A, a well-known factor driving muscle atrophy, was highly expressed in periodontitis-affected gingiva and was also elevated in the serum. Gingival overexpression of activin A via AAV-*Inhba* transduction activated canonical activin signaling in skeletal muscle, leading to significant muscle loss, whereas intra-gingival si*Inhba* injection to silence activin A expression in periodontitis-affected tissues prevented muscle loss. Importantly, we demonstrated that circulating activin A levels were significantly higher in periodontitis patients than in healthy individuals. Furthermore, older individuals with periodontitis exhibited significantly reduced grip strength compared with age-matched, periodontally healthy controls, thereby confirming the clinical relevance of our findings. Overall, these findings suggest that periodontitis is a potential risk factor for muscle atrophy and that inhibiting activin A expression in periodontal tissue could offer a therapeutic approach to prevent sarcopenia induced by periodontitis.

## Results

### Alveolar bone resorption pattern in ligature-induced periodontitis

The ligature-induced periodontitis (LIP) model rapidly induces alveolar bone loss in mice, with significant resorption occurring within 3 days and plateauing after 11 days[9]. To validate this, we used micro-CT to monitor alveolar bone changes over two weeks following ligature placement (Fig. 1a). Alveolar bone resorption was assessed by measuring the exposed tooth root length uncovered by alveolar bone, specifically the distance between the cementoenamel junction (CEJ) and the alveolar bone crest (ABC), using the palatal groove of the ligature-tied second molar as a reference point. Consistent with

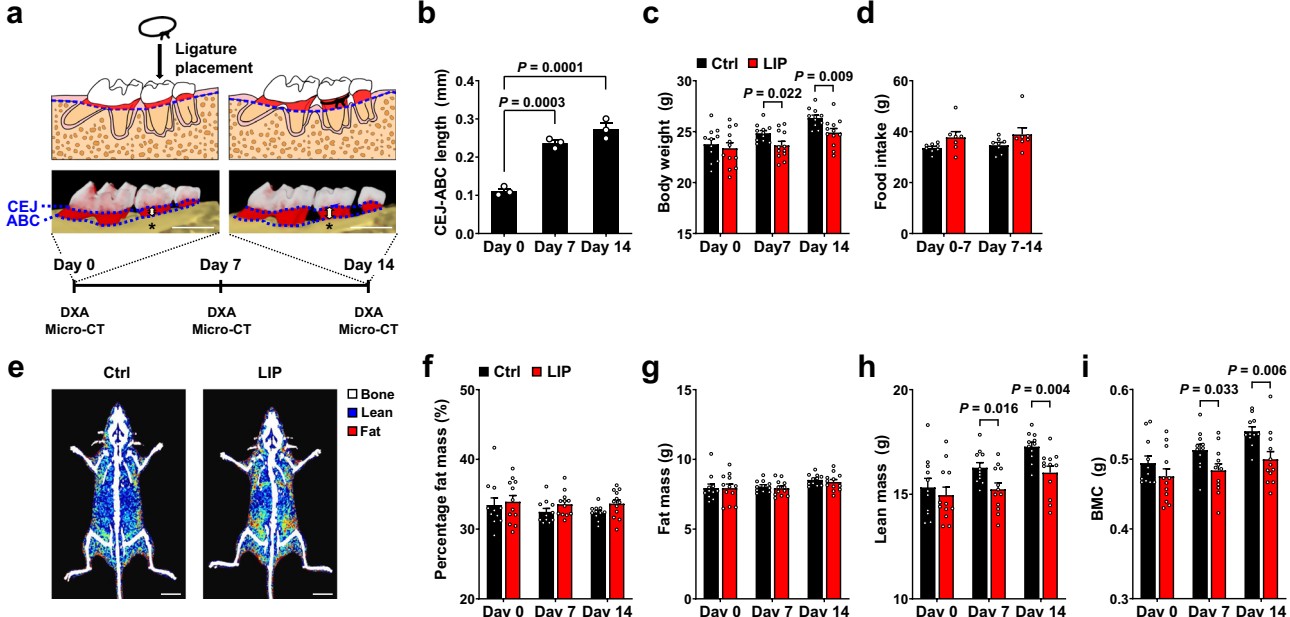

**Fig. 1 | A two-week ligature-induced periodontitis (LIP) model leads to alveolar bone loss and affects overall body composition. a** Timeline of the LIP model. Degree of alveolar bone loss was evaluated by measuring the distance between cementoenamel junction (CEJ) and alveolar bone crest (ABC) marked with double-headed arrows and asterisks in the representative micro-CT images. Scale bar, 1 mm. **b** Changes in CEJ-ABC length after ligature placement (*n* = 3 each). **c** Changes in body weight after ligature placement (Ctrl: *n* = 11, LIP: *n* = 12). **d** Measurement of food intake after ligature placement (Ctrl: *n* = 8, LIP: *n* = 7). **e** Representative DXA scans 14 days after ligature placement. Scale bar, 1 cm. **f–i** Changes in fat mass (**f,g**), lean mass (**h**), and bone mineral content (BMC) (**i**) after ligature placement (Ctrl: *n* = 11, LIP: *n* = 12). Data represent mean ± SEM. Statistical significance was assessed by one-way ANOVA with Tukey's post hoc test (**b**) or two-tailed Student's *t*-test (**c,d,f,g,h,i**). Source data are provided as a Source Data file.

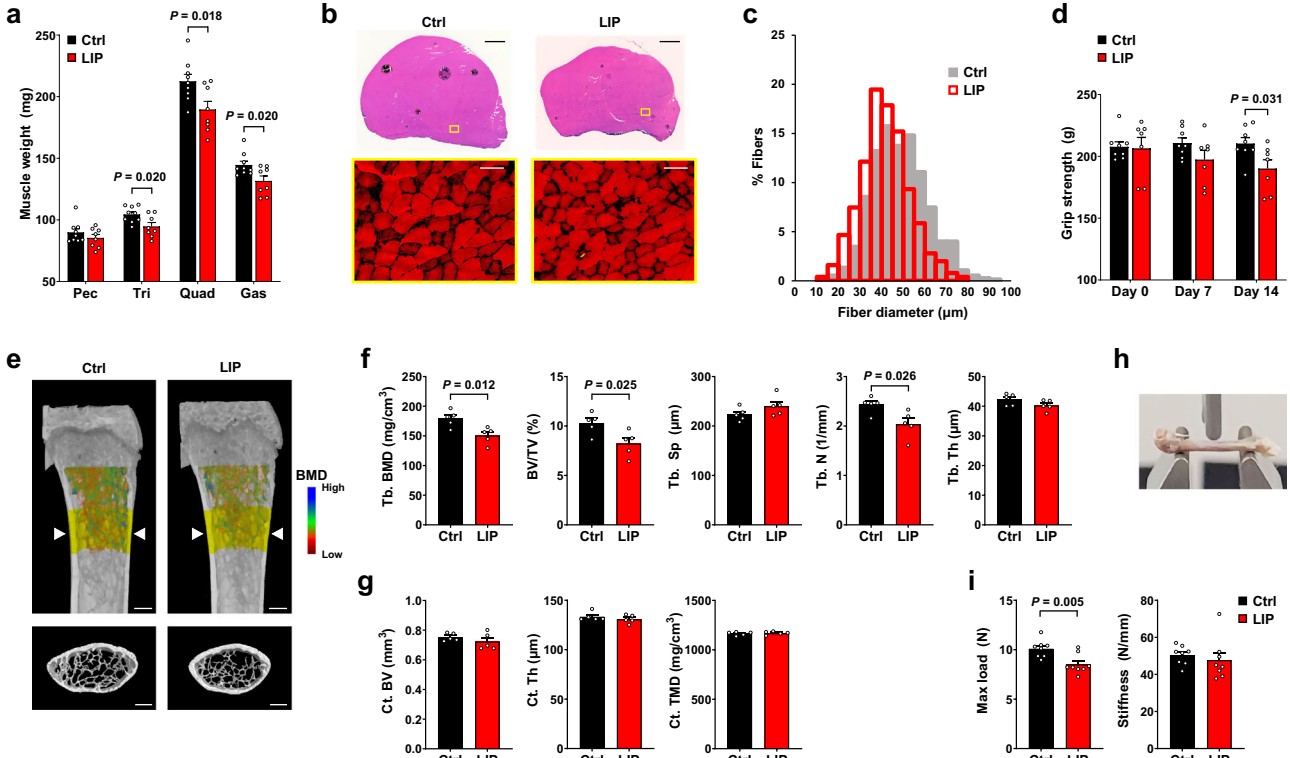

**Fig. 2 | Periodontitis induces systemic loss of muscle and trabecular bone.**
**a** Weights of the pectoralis (Pec), triceps (Tri), quadriceps (Quad), and gastrocnemius (Gas) muscles 14 days after ligature placement (Ctrl: $n = 9$, LIP: $n = 8$).
**b** Hematoxylin and eosin (H&E)-stained quadriceps sections 14 days after ligature placement. Magnified images of the yellow-boxed regions are shown in the lower panel. Scale bars: 1 mm (top), 50 µm (bottom). **c** Distribution of muscle fiber diameters in the quadriceps muscles shown in (**b**) ($n = 3$ each). Fiber diameters were plotted as a percentage of the total fiber number (750 fibers per group). Mean fiber diameters: $48.21 \pm 0.46$ µm (Ctrl) and $41.71 \pm 0.41$ µm (LIP). **d** Changes in grip strength after ligature placement (Ctrl: $n = 8$, LIP: $n = 7$). **e** Representative micro-CT images of the distal femur 14 days after ligature placement. Bone mineral density

(BMD) is indicated by color. The yellow region in the top panel indicates the area analyzed for cortical (Ct) bone. The transverse view images of the area marked by arrowheads are displayed in the bottom panel. Scale bar, 0.5 mm.
**f,g** Histomorphometric analysis of the micro-CT images of the trabecular (**f**) and cortical (**g**) bone of the distal femurs shown in (**e**) ($n = 5$ each). **h** Representative image of three-point bending test performed on the femur. **i** Mechanical properties of the femur 14 days after ligature placement ($n = 8$ each). Tb, Trabecular; BV/TV, Bone Volume/Total Volume; Tb. Sp, Tb. separation; Tb.N, Tb. number; Tb. Th, Tb. thickness; Ct. TMD, Ct. tissue mineral density. Data represent mean ± SEM. Statistical significance was assessed by two-tailed Student's $t$-test (**a,d,f,g,i**). Source data are provided as a Source Data file.

previous studies, significant bone resorption occurred within the first 7 days, followed by additional but non-significant bone loss over the next 7 days (Fig. 1b)[9–11]. These results align with previous findings that anaerobic bacterial counts peak on day 5, then decline by day 8[10]. Together, these observations suggest that the LIP induces active bone resorption within the first 7 days, followed by a gradual slowing of periodontitis progression over the next 7 days, leading to a subsequent stabilization phase.

### Body composition is altered in periodontitis independent of caloric intake
To evaluate the impact of periodontitis on the musculoskeletal system, we tracked body composition changes using dual-energy X-ray absorptiometry (DXA) for two weeks following ligature placement (Fig. 1a). DXA analysis revealed a gradual reduction in body weight in the LIP group compared with the control group (Fig. 1c). Since periodontitis can weaken masticatory function and potentially reduce food intake, we measured food consumption but found no significant difference between the groups (Fig. 1d). To exclude the possibility of malnutrition resulting from any potential impairment in masticatory efficiency despite normal food intake, we measured postprandial blood nutrient levels. We found no significant differences in total protein, albumin, total cholesterol, or triglycerides between the LIP and control mice (Supplementary Fig. 1a–d). Interestingly, postprandial blood glucose levels increased as periodontitis progressed (Supplementary Fig. 1e), which is

consistent with a previous report that periodontal inflammation negatively affects glycemic control[12]. While the LIP group showed a trend toward an increased percentage of fat mass (Fig. 1e, f), their absolute fat mass did not differ from that of the control (Fig. 1g). This indicates that the higher fat percentage in the LIP group reflects a decrease in total body weight (Fig. 1c) rather than an actual increase in fat mass. The unchanged absolute fat mass further suggests that reduced caloric intake was not the primary driver of weight loss. Unlike fat mass, lean mass and bone mineral content (BMC), which are predictive of muscle and bone mass, respectively, steadily decreased in the LIP group compared with the control, suggesting that periodontitis may contribute to musculoskeletal atrophy (Fig. 1h, i).

### Periodontitis induces systemic musculoskeletal atrophy
Since DXA measurements of lean mass include non-muscle tissues such as viscera, we conducted a more muscle-specific analysis by measuring the weights of the pectoralis, triceps, quadriceps, and gastrocnemius (Fig. 2a, Supplementary Fig. 2a). Muscle weights in the LIP group showed a downward trend in all muscles examined by day 7 and reached statistical significance in the triceps, quadriceps, and gastrocnemius by day 14. Histological analysis of each muscle at day 14 revealed an overall decrease in muscle size and a reduction in individual muscle fiber diameter (Fig. 2b, c, Supplementary Fig. 2b–g).
To determine whether periodontitis induces fiber type–specific effects, we performed immunostaining for myosin heavy chain

isoforms in the gastrocnemius muscle (Supplementary Fig. 2h, i). Fast-twitch type II fibers were more susceptible to atrophy than slow-twitch type I fibers, with the greatest diameter reduction in type IIb (22.5%), followed by type IIx (19.7%), type IIa (13.2%), and type I fibers (7.0%) (Supplementary Fig. 2j). This pattern parallels age-associated muscle atrophy, in which fast-twitch glycolytic fibers are more vulnerable than slow-twitch oxidative fibers[13,14]. Regarding fiber type composition, the proportion of type I fibers remained unchanged, whereas type IIb fibers showed a modest decrease with corresponding trends toward increased type IIa and type IIx fibers (Supplementary Fig. 2i, k), suggesting a potential shift toward more oxidative fiber types. However, these changes did not reach statistical significance and require further investigation. Finally, grip strength was significantly reduced in periodontitis-induced mice compared with controls (Fig. 2d), indicating that these structural changes translate into functional impairment.

While periodontitis reduced BMC (Fig. 1i), this does not necessarily confirm systemic bone loss, as BMC also reflects localized alveolar bone resorption caused by periodontitis. Therefore, to assess the systemic bone condition, we conducted additional analyses on the femur and vertebra. Micro-CT analysis of the distal femur revealed significant reductions in trabecular bone mineral density, bone volume/total volume of interest, and trabecular bone number in the LIP group (Fig. 2e, f). A similar decline in trabecular bone was observed in the lumbar spine, although it was not statistically significant (Supplementary Fig. 2l, m). Femoral cortical bone showed no differences between groups (Fig. 2e, g), suggesting that cortical bone may require longer periodontitis exposure or that periodontitis-induced bone loss is specific to trabecular bone. To determine whether reduced bone mass compromises biomechanical strength, we performed three-point bending tests on the femur (Fig. 2h). Femurs from periodontitis-induced mice exhibited significantly reduced maximal load and a trend toward decreased stiffness compared with controls (Fig. 2i), indicating compromised bone strength.

**Periodontitis induces gingival *Inhba* expression, elevates systemic activin A levels, and activates activin signaling in muscle**
Next, we investigated the mechanism underlying periodontitis-induced musculoskeletal atrophy. To identify genes responsible for the systemic secretion of musculoskeletal atrophy-inducing factors, we analyzed RNA-seq data from the gingiva of periodontitis-affected mice[15] (Fig. 3a). We applied three criteria to identify candidate genes. First, the gene must encode a secretion-related protein, as indicated by the GO term "secretion". Second, the gene must show at least a 4-fold increase in expression in the periodontitis group compared to controls, with a false discovery rate (FDR) below 0.01. Third, since low protein levels are unlikely to exert systemic effects, the gene should rank within the top 25% of expressed genes in periodontitis. Seven genes (*Mmp13*, *Spp1*, *S100a8*, *S100a9*, *Igf1*, *Mctp1*, and *Inhba*) met these criteria, most of which are well-documented for their local roles in periodontitis. Interestingly, although it has been less studied in the context of periodontitis, *Inhba* (which encodes activin A) is well known for its role in systemic muscle atrophy in conditions such as cancer cachexia[16,17].

To determine whether *Inhba* expression increases in the gingiva during periodontitis, we performed quantitative reverse transcription polymerase chain reaction (qRT-PCR) analysis on healthy gingiva and gingiva one day after periodontitis induction. Additionally, to assess whether gingival *Inhba* expression is high enough to exert systemic effects, we analyzed its expression in the quadriceps muscle, which undergoes atrophy in periodontitis, and in the liver, which has been reported as the tissue with the highest *Inhba* expression[18] (Fig. 3b). qRT-PCR analysis revealed that *Inhba* expression level in periodontitis-induced gingiva increased nearly 10-fold compared to healthy gingiva, was over 40-fold higher than in the quadriceps muscle, and was significantly higher than in the liver, which is considered the primary

*Inhba*-expressing organ. Since *Inhba* can heterodimerize with *Inhbb* or *Inha* to form activin AB or inhibin A, respectively, rather than forming activin A as a homodimer, we examined the expression of *Inhbb* and *Inha* to determine whether elevated *Inhba* might contribute to the upregulation of activin AB or inhibin A instead of activin A. While *Inhbb* expression was higher in the gingiva than in other tissues, its levels remained unchanged in periodontitis. In contrast, *Inha* expression significantly decreased in periodontitis-induced gingiva compared to healthy gingiva. These results suggest that *Inhba* upregulation in periodontitis primarily leads to increased activin A production, and the fact that its expression level is higher than in the liver indicates that periodontitis-induced activin A upregulation may exert systemic effects.

To examine how *Inhba* expression changes in the gingiva during the progression of periodontitis, we performed qRT-PCR to analyze the expression of activin and inhibin subunit genes, along with genes associated with tissue destruction (*Ctsk*, *Mmp9*, and *Mmp13*) and inflammation (*Il1b*, *Il6*, and *Tnf*) (Fig. 3c). Most tissue destruction and inflammation-related genes showed a significant increase on day 1, remained elevated until day 7, and declined by day 14, consistent with previous findings[11]. This pattern corresponds with rapid alveolar bone loss observed until day 7, followed by a slowing of resorption by day 14 (Fig. 1b). *Inhba* followed a similar trend, with expression sharply increasing from day 1 and declining by day 14. Western blot analysis confirmed this increase at the protein level (Supplementary Fig. 3a). However, unlike *Inhba, Inhbb* expression remained stable, and *Inha* expression showed a downward trend.

To determine whether gingival *Inhba* expression leads to systemic activin A circulation, we measured serum activin A levels using enzyme-linked immunosorbent assay (ELISA) (Fig. 3d). Serum activin A levels significantly increased as early as day 1 after periodontitis induction, remained elevated until day 7, and returned to pre-induction levels by day 14. This timeline corresponds to the decline in periodontitis progression, as indicated by the decreased rate of alveolar bone loss observed on day 14 (Fig. 1b). During the first week, when serum activin A levels were elevated, specifically on days 3 and 7 after periodontitis induction, we observed increased levels of phosphorylated SMAD3 (pSMAD3) in both the quadriceps and extensor digitorum longus (EDL) muscles, along with upregulated expression of atrophy-related genes (*Foxo3*, *Fbxo32*, and *Trim63*) and MuRF1 protein (encoded by *Trim63*) (Fig. 3e–g, Supplementary Fig. 3b–d). These findings indicate that the canonical activin A signaling pathway was activated during the active phase of periodontitis, leading to muscle atrophy.

**Gingiva-derived activin A is sufficient to induce muscle atrophy**
Given that periodontitis elevates systemic activin A levels along with other inflammatory mediators, we sought to determine whether gingiva-derived activin A alone is sufficient to induce muscle atrophy. To achieve localized overexpression specifically within gingival tissue, we utilized an AAV-mediated expression system delivered via direct intra-gingival injection (Fig. 3h). The injection volume was carefully optimized to ensure localized expression and minimize the diffusion of viral particles to adjacent tissues (Supplementary Fig. 3e). Based on the SMAD-responsive luciferase reporter assay, which showed that larger tags interfered with activin A bioactivity (Supplementary Fig. 3f, g), we used untagged AAV-*Inhba* for functional studies and AAV-*Inhba-FLAG* for protein tracking, with AAV-*EGFP* as the control vector in all experiments.

To determine whether gingiva-derived activin A reaches skeletal muscle, we transduced the gingiva with AAV-*Inhba-FLAG* (Supplementary Fig. 3h). Immunofluorescence (IF) analysis revealed FLAG signals on the quadriceps muscle membrane, colocalizing with activin type II receptors (Fig. 3i). Gingival AAV-*Inhba* injection elevated serum activin A levels, increased pSMAD3, and upregulated atrophy-related

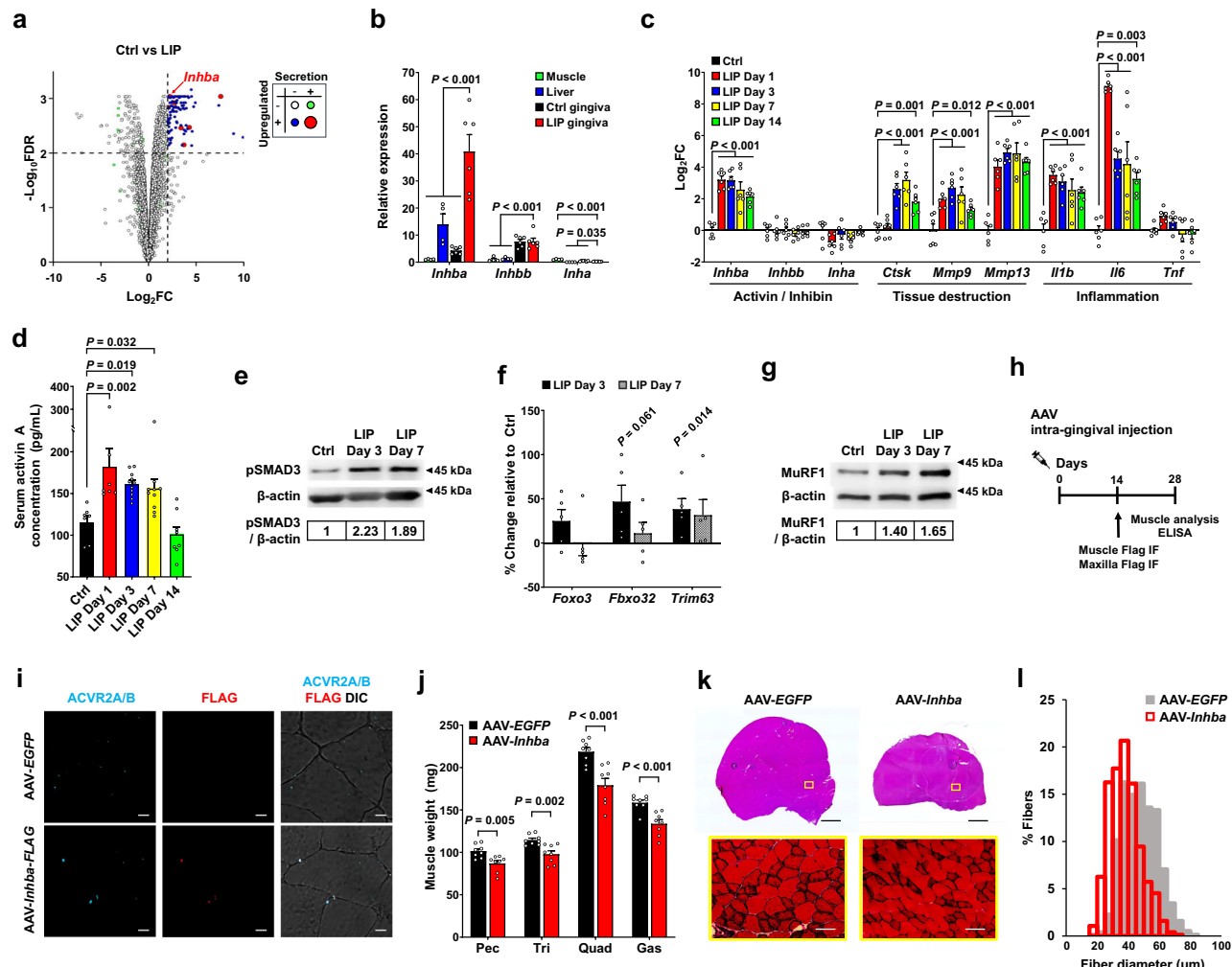

**Fig. 3 | Periodontitis upregulates gingival activin A expression, elevates systemic activin A levels, and activates activin signaling in muscle. a** Volcano plot of differentially expressed genes in mouse gingiva (Ctrl vs. LIP; GSE186882). Horizontal and vertical dashed lines indicate significance (FDR ≤ 0.01) and $\log_2$FC ≥ 2, respectively. Only the top 25% of highly expressed genes are plotted. **b** qRT-PCR of mouse muscle, liver, and gingiva (healthy vs. periodontitis-induced). Expression is relative to muscle (muscle and liver: $n = 4$; gingiva: $n = 6$ per group). **c** qRT-PCR of mouse gingiva ($n = 6$ each). Unlike *Inhba*, other activin and inhibin subunit genes remained stable. **d** Serum activin A levels during periodontitis progression (Ctrl and LIP day 1: $n = 7$, LIP day 3: $n = 11$, LIP day 7: $n = 10$, LIP day 14: $n = 8$). **e** Representative Western blot of pSMAD3 in quadriceps muscles. **f** Percent change in expression of muscle atrophy-related genes in quadriceps muscles ($n = 5$ each). **g** Representative Western blot of MuRF1 in quadriceps muscles. **h** Intra-gingival AAV injection

scheme. **i** Immunofluorescence staining for FLAG and activin type II receptors (ACVR2A/B). FLAG colocalizes with ACVR2A/B in quadriceps of intra-gingivally AAV-injected mice. Scale bar, 10 μm. **j** Weights of Pec, Tri, Quad, and Gas muscles 4 weeks after intra-gingival AAV-*EGFP* or AAV-*Inhba* transduction ($n = 8$ each). **k** H&E-stained quadriceps sections 4 weeks post-transduction. Lower panels show magnified images of yellow-boxed regions. Scale bars: 1 mm (top), 50 μm (bottom). **l** Distribution of quadriceps muscle fiber diameters from (**k**) ($n = 3$ each). Mean fiber diameters: 47.47 ± 0.39 μm (AAV-*EGFP*) and 37.89 ± 0.36 μm (AAV-*Inhba*). Data (**b,c,d,f,j**) represent mean ± SEM. Significance was assessed by one-way ANOVA with Holm-Šídák post hoc test (**b–d**) or two-tailed Student's *t*-test (**f, j**). Comparisons were made against the LIP group (**b**) or Ctrl group (**c, d**). Source data and exact *P* values are provided as a Source Data file.

genes and MuRF1 protein in both the quadriceps and EDL muscles (Supplementary Fig. 3i–o). This was accompanied by significant reductions in muscle mass and myofiber diameter (Fig. 3j–l), demonstrating that gingiva-derived activin A is sufficient to drive skeletal muscle atrophy independent of other inflammatory signals.

## Fibroblasts, myeloid cells, and proliferating epithelial cells are major *Inhba*-expressing cell populations

To identify the cellular sources of activin A in periodontitis, we performed single-cell RNA-seq (scRNA-seq) on gingival tissues from control and ligature-induced periodontitis mice ($n = 3$ per group; 88,191 cells; Fig. 4a). Eight major cell populations were identified, with fibroblasts, immune cells, junctional epithelial cells (JE), and pericytes representing the primary *Inhba*-expressing cell populations (Fig. 4b, Supplementary Fig. 4a, b). Among immune cells, myeloid cells were the

predominant source. Within the JE, *Inhba* expression was markedly elevated in a cycling epithelial cluster, predominantly in the S/G2/M phase, upon periodontitis induction (Fig. 4c, Supplementary Fig. 4c). Similarly, although gingival and sulcular epithelial cells (GSE) showed low average *Inhba* expression, *Inhba* levels were significantly elevated within their cycling subset upon periodontitis induction (Fig. 4c, Supplementary Fig. 4d), indicating that epithelial *Inhba* production is primarily concentrated in the proliferating basal layer. Overall, *Inhba* expression significantly increased in fibroblasts, myeloid cells, cycling JE, and cycling GSE during periodontitis (Fig. 4c, d). These findings reveal a spatially compartmentalized *Inhba* production during periodontitis: the proliferating basal epithelium serves as the principal epithelial source, while fibroblasts and infiltrating myeloid cells constitute the major sources within the connective tissue (Fig. 4e). Additionally, we mapped the cellular origins of other potential systemic

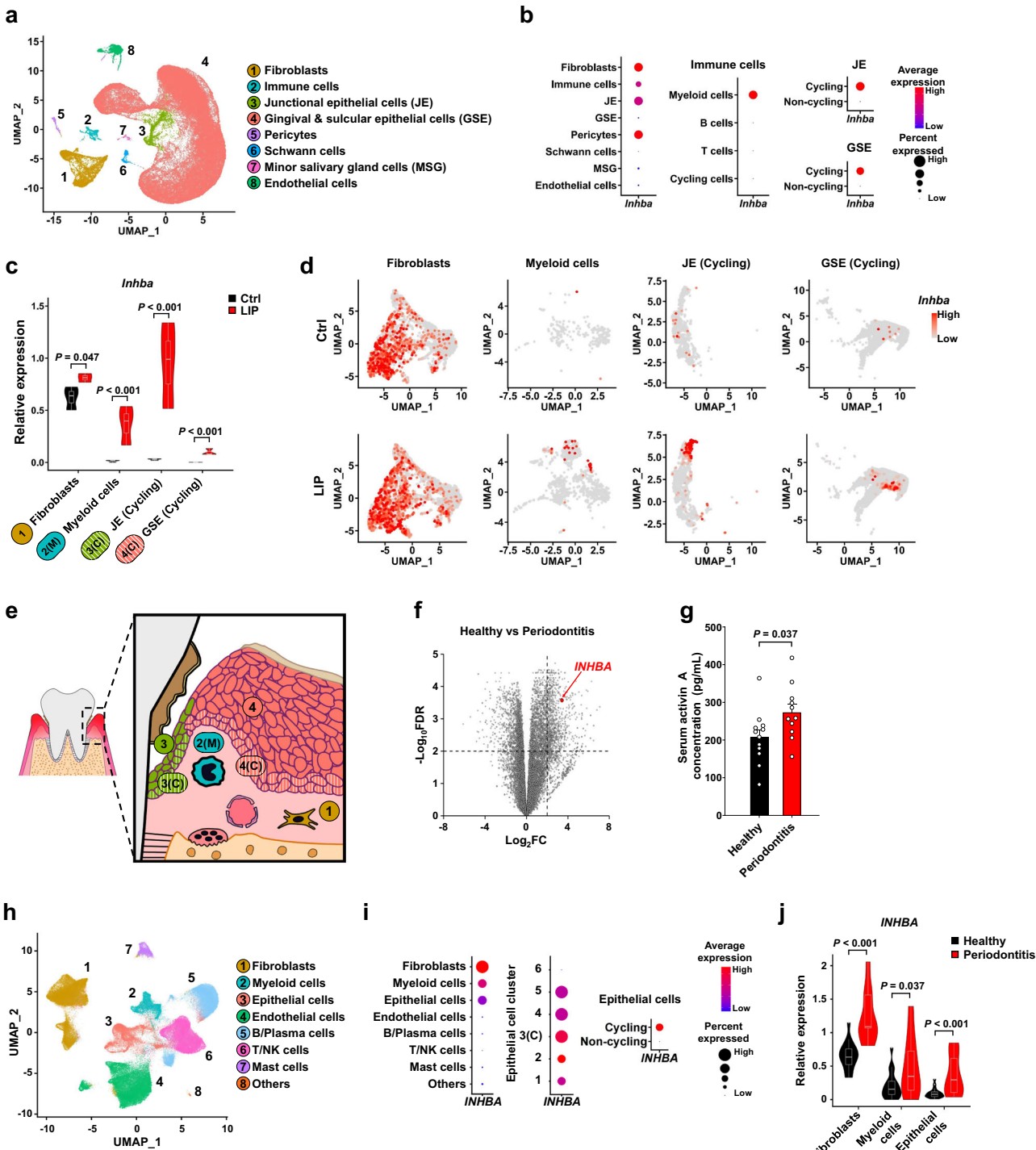

**Fig. 4 | Identification of activin A-producing cell populations in mouse and human gingiva during periodontitis. a** Uniform Manifold Approximation and Projection (UMAP) plot of mouse gingiva, integrated from control and periodontitis-induced datasets (*n* = 3 each, GSE315263). **b** Dot plots of *Inhba* expression across annotated cell types. **c** Relative *Inhba* expression in fibroblasts, myeloid cells, and cycling epithelial cells (JE and GSE) (*n* = 3 each). **d** Feature plots visualizing *Inhba* expression in fibroblasts, myeloid cells, and cycling epithelial cells (JE and GSE). **e** Schematic diagram showing the spatial localization of cell populations with significantly upregulated *Inhba* expression following periodontitis induction. **f** Volcano plot of differentially expressed genes in the gingiva of healthy human individuals versus individuals with periodontitis (GSE223924). Horizontal and vertical dashed lines indicate significance (FDR ≤ 0.01) and log$_2$FC ≥ 2, respectively. **g** Serum activin A concentrations in healthy controls and periodontitis patients with alveolar bone loss extending to the mid-third of the root and beyond (Healthy: *n* = 12, Periodontitis: *n* = 11). **h** UMAP plot of the human gingiva from integrated datasets composed of healthy and periodontitis-affected gingiva (GSE152042 and GSE164241). **i** Dot plots of *INHBA* expression across annotated cell types. **j** *INHBA* relative expression level in fibroblasts, myeloid cells, and epithelial cells (Healthy: *n* = 15, Periodontitis: *n* = 9). Data (**g**) represent mean ± SEM. Box plots (**c,j**) show the median (center line), interquartile range (IQR; box bounds: 25th to 75th percentile), 1.5x IQR (whiskers), and outliers as individual points. Data (**c,j**) were calculated by pseudobulking the dataset and the graphs are presented as a combination of violin and box plots. Statistical significance was assessed by two-tailed Student's *t*-test (**g**), or Wald test adjusted by Benjamini-Hochberg correction for multiple comparisons in DESeq2 (**c, j**). Demographics of human subjects (**f–j**) are provided in Supplementary Table 3, 4, and 5. Source data and exact *P* values are provided as a Source Data file.

mediators—*Mmp13*, *S100a8*, *S100a9*, *Igf1*, *Spp1*, and *Mctp1* (Supplementary Fig. 4e).

## Systemic activin A is elevated in human periodontitis, with conservation of *INHBA*-expressing cells across species

To explore whether a similar mechanism occurs in humans, we analyzed bulk RNA-seq data from gingival tissues of individuals with periodontitis[19] (Fig. 4f). *INHBA* expression was significantly elevated in periodontitis-affected human gingiva, exhibiting a ~10-fold increase (Fig. 4f), which exceeded the ~4-fold increase observed in mouse gingiva (Fig. 3a). We next measured circulating activin A levels in periodontitis patients with alveolar bone loss extending to the mid-third of the root and beyond, and found significantly elevated serum levels compared with healthy controls (Fig. 4g). Furthermore, analysis of data from the Korea National Health and Nutrition Examination Survey (KNHANES) revealed that periodontitis was associated with significantly reduced grip strength among age-matched older adults (Supplementary Table 1, 2). This is consistent with longitudinal observations linking periodontal disease to an accelerated decline in muscle function[8]. Together, these results support the translational relevance of our murine model and suggest that periodontitis-induced activin A elevation may contribute to impaired skeletal muscle function in humans.

To identify the major cellular sources of *INHBA* in human periodontitis, we analyzed publicly available scRNA-seq datasets from human gingival tissues[20,21] (Fig. 4h). Fibroblasts, myeloid cells, and epithelial cells were the primary *INHBA*-expressing populations (Fig. 4i, Supplementary Fig. 5a–g). Epithelial cells were further classified into six subclusters, among which clutser 3 (C3) exhibited the highest overall *INHBA* expression, based on both expression levels and the percentage of expressing cells (Fig. 4i, Supplementary Fig. 5h, i). Notably, C3 also displayed the highest proliferative activity, suggesting its localization within the basal cell layer. Periodontitis significantly upregulated *INHBA* expression in fibroblasts, myeloid cells, and epithelial cells (Fig. 4j), a trend further supported by feature plots showing a marked increase in *INHBA* expression across these populations (Supplementary Fig. 5j). These findings establish that *INHBA* upregulation in periodontitis is driven by cellular mechanisms conserved between human and mice, supporting activin A as a therapeutically relevant target for preventing periodontitis-associated muscle atrophy.

## Tissue remodeling and gene expression changes contribute to increased activin A levels

To investigate the spatial pattern of *Inhba* expression in periodontal tissue, we conducted RNA fluorescence in situ hybridization (FISH) and immunofluorescence (IF) staining on gingival samples (Fig. 5a–f). Unlike previous setups, where periodontitis was induced bilaterally in the LIP group (Figs. 1, 2, 3a–g, 4a–d), a unilateral ligature was applied to enable a more objective comparison of *Inhba* expression between the ligatured side and the control side on a single microscope slide. *Inhba* RNA FISH and Ki-67 IF staining revealed a dramatic increase in *Inhba* expression and cell proliferation in the gingiva on the ligatured side compared to the untreated control side (Fig. 5b). Additionally, CD11b IF staining, a myeloid cell marker, confirmed extensive myeloid cell infiltration in periodontitis-induced gingiva (Fig. 5c).

Observation of the outer epithelium facing the oral cavity showed thickening of the epithelial layer, likely due to hyperplasia driven by increased cell proliferation, as indicated by elevated Ki-67 expression (Fig. 5d). Ki-67 was predominantly expressed in the basal layer, which also showed a notable increase in *Inhba* expression. Under periodontitis induction, *Inhba* expression increased in basal layer cells, which underwent active proliferation, leading to an increased number of *Inhba*-expressing cells and a substantial overall increase in *Inhba* expression. The inner epithelium, composed of junctional and sulcular

epithelium and facing the tooth, also exhibited active cell division in the basal layer upon periodontitis, as indicated by Ki-67 expression (Fig. 5e). Unlike the outer epithelium, resident myeloid cells were observed in the inner epithelium. Under healthy conditions, these cells exhibited low *Inhba* expression but became activated in response to periodontitis, leading to increased *Inhba* expression. Additionally, during periodontitis, increased infiltration of activated myeloid cells into the inner epithelium, particularly the surface layer, was detected, further enhancing *Inhba* expression in this region. Periodontitis also induced significant hyperplasia and myeloid cell infiltration in the connective tissue, as indicated by increased Ki-67 and CD11b staining (Fig. 5f). Most connective tissue cells expressed *Inhba*, with particularly high expression in multinucleated CD11b-positive osteoclasts near the alveolar bone. Due to their large size, osteoclasts are typically excluded from scRNA-seq analysis, leaving their in vivo gene expression largely uncharacterized. Here, we utilized *Inhba* RNA FISH to reveal that osteoclasts express *Inhba*, potentially contributing to systemic muscle atrophy. However, during in vitro osteoclast differentiation, from myeloid precursor cells to fully differentiated, functional osteoclasts, *Inhba* expression was highest at early stages and progressively decreased as osteoclastogenesis proceeded (Supplementary Fig. 5k–m). This suggests that the *Inhba* signal detected in osteoclasts likely reflects the high baseline expression of their myeloid precursor cells. Nevertheless, the spatial localization of *Inhba*-expressing osteoclast-lineage cells at sites of active bone resorption implies a potential contribution of the myeloid lineage to both local bone loss through osteoclastogenesis and systemic muscle loss through activin A secretion (Fig. 5g).

## Gingival *Inhba* silencing or systemic activin A blockade prevents periodontitis-induced skeletal muscle atrophy

A previous study demonstrated that injecting tumor-targeting si*Inhba* into an orthotopic pancreatic ductal adenocarcinoma mouse model prevented muscle atrophy[22]. Building on this approach, we locally injected si*Inhba* into the gingiva of periodontitis-affected mice to assess whether si*Inhba* injection could influence the systemic loss of muscle and bone induced by periodontitis (Fig. 6a). Given that serum activin A levels remain elevated at least until day 7 (Fig. 3d), we administered si*Inhba* a total of 3 times at 3-day intervals over 6 days, starting immediately after periodontitis induction.

The si*Inhba* injections significantly and specifically reduced *Inhba* expression in the gingiva without affecting *Inhbb* or *Inha* (Fig. 6b, Supplementary Fig. 6a). On day 7, when serum activin A levels were markedly elevated in periodontitis-affected mice, si*Inhba* treatment significantly reduced activin A levels (Fig. 6c). Correspondingly, pSMAD3 levels in the quadriceps muscle were also reduced (Supplementary Fig. 6b). Serum activin A levels of periodontitis-induced mice had returned to baseline by day 14, likely because gingival *Inhba* expression no longer influenced systemic activin A circulation, and, as expected, si*Inhba* injection into the gingival tissue did not significantly reduce serum activin A levels on day 14. Because activin A expressed in the gingival tissue during the active phase of periodontitis (approximately 7 days in the LIP model) contributes to the elevation of serum activin A levels, administering si*Inhba* into the periodontitis-affected gingiva during this phase would be beneficial to effectively prevent the increase in serum activin A levels.

To assess whether si*Inhba* injections affect periodontitis progression, we performed micro-CT analysis of the alveolar bone (Fig. 6d, e). There were no significant differences in alveolar bone loss, as measured by CEJ-ABC lengths, between groups, indicating that si*Inhba* injections did not suppress periodontitis progression. Although the expression of genes related to tissue destruction and inflammation in the gingiva tended to decrease in the LIP-si*Inhba* group compared to the LIP-siCtrl group, most changes were not statistically significant except for *Mmp13* and *Il1b* (Supplementary Fig. 6c). The fact that

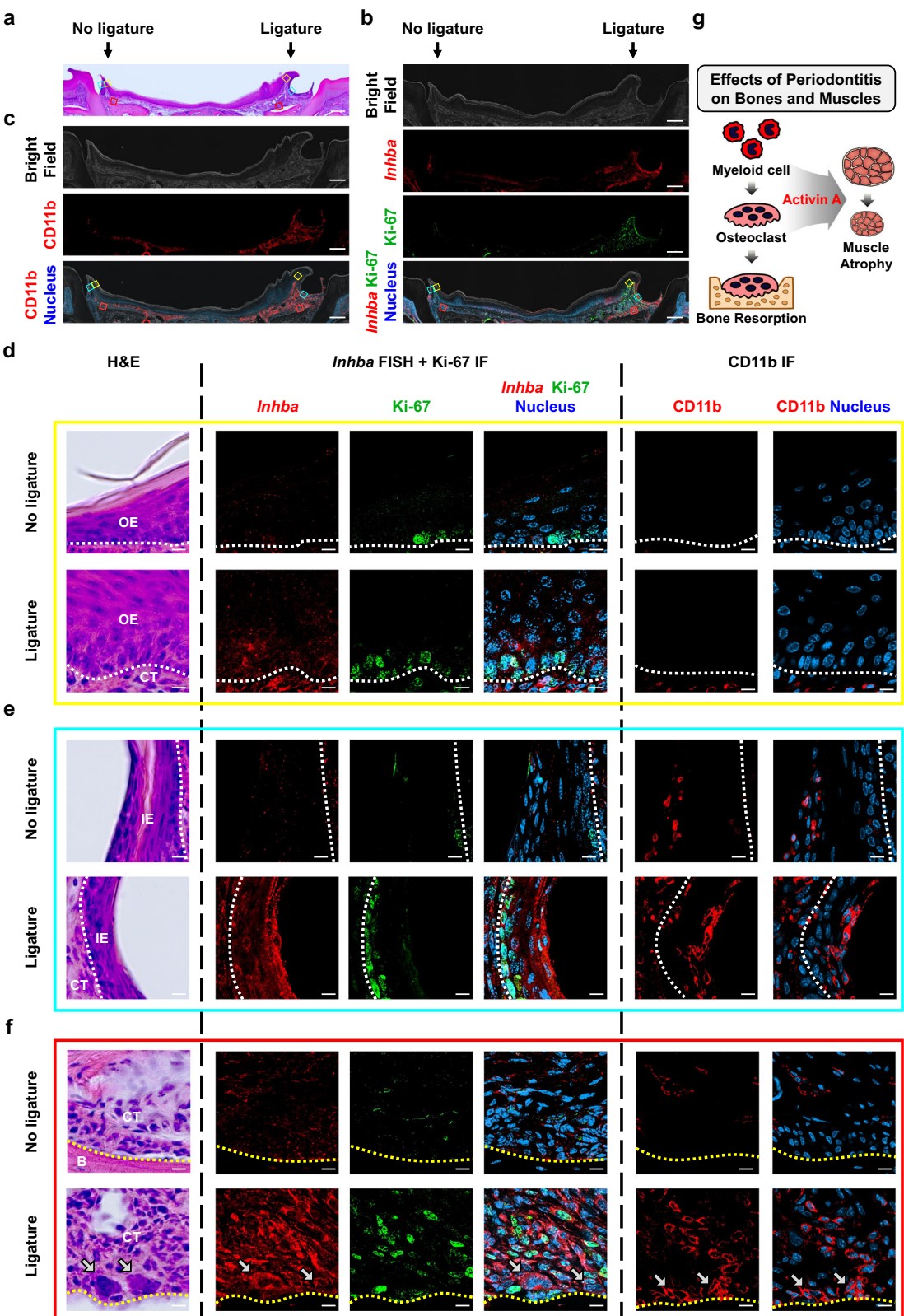

**Fig. 5 | Tissue remodeling and changes in gene expression patterns observed in periodontitis lead to the upregulation of activin A. a** H&E staining of mouse periodontal tissue (unilateral ligature-induced periodontitis). Scale bar, 0.2 mm. **b, c** *Inhba* RNA fluorescence in situ hybridization (FISH) and Ki-67 immunofluorescence (IF) (**b**), and CD11b IF (**c**) on adjacent sections. Boxed regions (yellow, sky blue, red) in (**a**–**c**) are magnified in (**d**–**f**). Scale bar, 0.2 mm. **d**–**f** In periodontitis, Ki-67 and *Inhba* are enriched in the basal layer of the oral epithelium (OE) (**d**). CD11b-positive myeloid cells with increased *Inhba* expression infiltrate the inner epithelium (IE), including the junctional and sulcular epithelium, particularly at the surface layer (**e**). In connective tissue (CT), marked hyperplasia and myeloid cell infiltration are observed; notably, multinucleated CD11b-positive osteoclasts (white arrows) adjacent to the alveolar bone (B) exhibit high *Inhba* expression (**f**). Dashed lines: epithelium-CT (white) and CT-B (yellow) boundaries. Scale bar, 10 μm. **g** Schematic of myeloid cell and osteoclast roles in periodontitis-induced muscle atrophy.

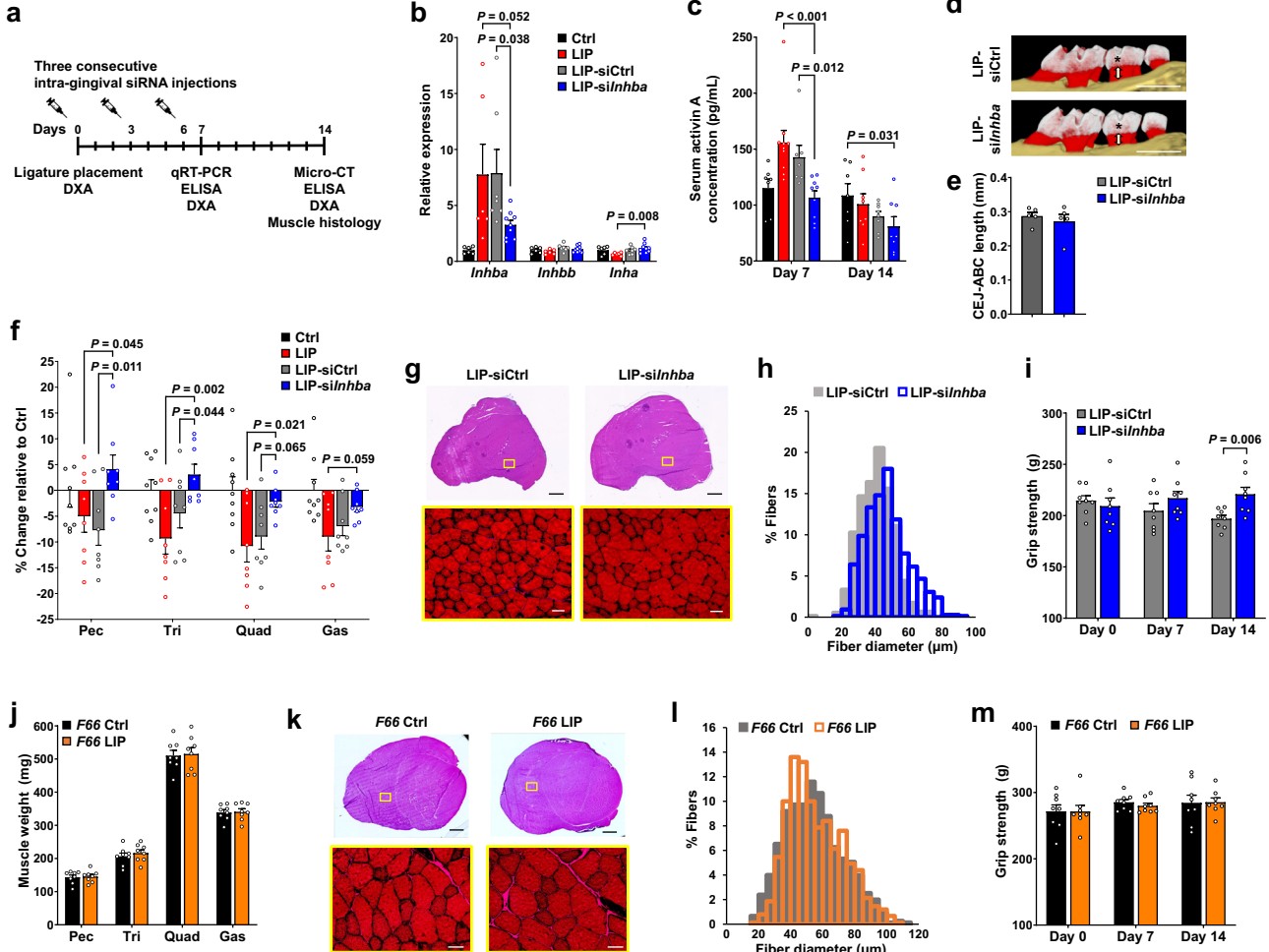

**Fig. 6 | Inhibition of activin A production or signaling prevents periodontitis-induced skeletal muscle atrophy. a** Intra-gingival siRNA injection scheme. **b** qRT-PCR analysis in mouse gingiva 7 days after ligature placement (Ctrl and LIP: *n* = 6, LIP-si*Inhba*: *n* = 7, LIP-si*Inhba*: *n* = 9). Data of the Ctrl and LIP groups are from Fig. 3c. **c** Serum activin A concentration. (On day 7: Ctrl, *n* = 7; LIP, *n* = 10; LIP-siCtrl, *n* = 7; LIP-si*Inhba*, *n* = 9. On day 14: Ctrl, *n* = 7; LIP, LIP-siCtrl, LIP-si*Inhba*, *n* = 8. Data of the LIP group are from Fig. 3d. **d,e** Representative micro-CT images (**d**) and bone loss quantification (**e**) 14 days after ligature placement (*n* = 5 each). Scale bar, 1 mm. **f** Percentage change in muscle weights 14 days after ligature placement (Ctrl: *n* = 9; LIP, LIP-siCtrl, LIP-si*Inhba*: *n* = 8). Data of the Ctrl and LIP groups are from Fig. 2a. **g,k** H&E-stained quadriceps sections 14 days after ligature placement. Magnified

images of the yellow-boxed regions are shown in the lower panel. Scale bars: 1 mm (top), 50 μm (bottom). **h,l** Distribution of muscle fiber diameters in the quadriceps muscles shown in (**g,k**) (*n* = 3 each). Mean fiber diameters: 41.12 ± 0.36 μm (LIP-siCtrl), 47.35 ± 0.46 μm (LIP-si*Inhba*), 54.67 ± 0.65 μm (*F66* control), and 55.74 ± 0.62 μm (*F66* LIP). **i,m** Changes in grip strength (*n* = 8 each). **j** Muscle weights in follistatin (*F66*) transgenic mice 14 days after ligature placement (*n* = 8 each). Data represent mean ± SEM. Statistical significance was assessed by one-way ANOVA with Fisher's LSD post hoc test for comparisons between the LIP-si*Inhba* group and other groups (**b,c,f**) or two-tailed Student's *t*-test (**e,i,j,m**). Source data and exact *P* values are provided as a Source Data file.

si*Inhba* injections reduced serum activin A levels without suppressing periodontitis suggests that elevated systemic activin A originates primarily from activin A produced in the periodontitis-affected gingiva, rather than as a secondary response to the disease.

DXA analysis showed a trend toward increased body weight and lean mass in the si*Inhba*-injected group compared to the siCtrl-injected group, although the increase in BMC was minimal, and fat mass remained unchanged (Supplementary Fig. 6d–g). To assess whether si*Inhba* injection rescues periodontitis-induced muscle atrophy, we measured the weights of the pectoralis, triceps, quadriceps, and gastrocnemius muscles and analyzed the percentage change relative to the controls (Fig. 6f). The LIP-si*Inhba* group showed increased muscle mass in all muscles compared to both the LIP without injection and LIP-siCtrl groups, with statistically significant gains in the pectoralis and triceps. Histological analysis further supported these findings, showing a larger overall muscle size and increased muscle fiber diameters in the LIP-si*Inhba* group compared to the LIP-siCtrl group (Fig. 6g, h). As previously shown in Fig. 2b, c, the LIP group exhibited a 13.5% decrease

in quadriceps muscle fiber diameter (from 48.21 to 41.71 μm) compared to the control group, whereas the LIP-si*Inhba* group showed a 15.2% increase (from 41.12 to 47.35 μm) compared to the LIP-siCtrl group. Grip strength was also significantly improved in the LIP-si*Inhba* group (Fig. 6i). This indicates that si*Inhba* injections effectively mitigated periodontitis-induced muscle atrophy and suggests that activin A is a key driver of muscle loss in periodontitis. However, unlike its effects on muscle, si*Inhba* injections had no significant impact on bone (Supplementary Fig. 6h–m). Micro-CT analysis of the distal femurs and vertebrae revealed no significant differences in trabecular or cortical bone between groups, and three-point bending tests on the femur showed only modest, non-significant trends toward increased maximum load and stiffness, indicating that systemic bone loss is less influenced by circulating activin A.

Having shown that reducing activin A production prevents muscle loss, we next asked whether blocking activin A signaling would be similarly protective. We employed *F66* transgenic mice, which overexpress follistatin, an endogenous inhibitor of activin A, specifically in

skeletal muscle. Following periodontitis induction, *F66* mice maintained normal body weight, body composition, muscle mass, myofiber diameter, and grip strength (Fig. 6j–m, Supplementary Fig. 6n–q). These results demonstrate that suppressing activin A signaling within muscle is sufficient to prevent periodontitis-induced muscle atrophy, despite ongoing periodontal inflammation, establishing activin A signaling as the essential pathway linking periodontitis to systemic muscle loss.

## Discussion

Activin A is a potent negative regulator of muscle mass and a key contributor to muscle wasting in cancer cachexia[16,17,23]. It has also been reported that activin A induces systemic muscle loss in chronic inflammatory diseases such as COPD and CKD[24,25]. In this study, we discovered that periodontitis systemically leads to muscle atrophy in mice and that activin A is elevated in periodontitis-affected tissues. Activin A was detected at high levels in the serum of both periodontitis-affected mice and humans. Using an AAV expression system, we further demonstrated that activin A overexpressed in the gingiva enters the circulation, binds to activin receptors in muscle, and activates canonical activin signaling, leading to significant muscle loss. Epidemiologic analysis further revealed reduced muscle function in older adults with periodontitis, confirming the clinical relevance of these findings.

Periodontitis-induced muscle atrophy was most pronounced in the quadriceps. Given that the quadriceps muscles are predominantly composed of fast-twitch fibers, a fiber-type-specific response was anticipated. Indeed, our analysis of the gastrocnemius, which contains a diverse mixture of fiber types, confirmed that this atrophy was significantly greater in fast-twitch fibers than in slow-twitch fibers. Since age-related muscle loss also preferentially affects fast-twitch fibers[13,14], the high prevalence of periodontitis among adults over 65 (approximately 60%)[26] suggests that periodontitis may serve as a significant, yet underrecognized, contributor to age-related muscle loss. The varying degrees of atrophy observed across different muscle groups may also reflect muscle-specific responsiveness to activin signaling, as prior studies have reported muscle-to-muscle variability in activin receptor expression[27,28].

Inflammatory conditions are known to upregulate activin A expression. In rheumatoid arthritis, fibroblast-like synoviocytes serve as key sources of activin A[29], while in CKD, activin A is primarily produced by fibroblasts and juxtaglomerular cells[24]. Additionally, LPS challenges have been shown to rapidly elevate serum activin A levels through activation of peripheral blood monocytes[30,31]. However, the relationship between periodontitis and activin A, particularly the specific cell types within the periodontium responsible for its production, remained unclear. Our findings reveal that fibroblasts, myeloid cells, and epithelial cells in the periodontium are principal sources of activin A, and that its expression increases significantly in response to periodontitis. Spatial analysis of *Inhba*-expressing cells identified strong signals in osteoclast-lineage cells near sites of bone resorption; however, these signals likely reflect high baseline *Inhba* expression inherited from myeloid precursors. Nevertheless, the presence of *Inhba*-expressing osteoclast-lineage cells on the bone surface suggests that myeloid cells not only contribute to systemic muscle loss via activin A secretion but also promote local bone resorption through osteoclast differentiation.

Beyond its well-established role in skeletal muscle, activin A regulates multiple non-muscle tissues. Elevated activin A signaling promotes cardiac and renal fibrosis and induces hepatocyte apoptosis[32]. These findings suggest that increased circulating activin A in periodontitis may affect multiple tissues and organs beyond skeletal muscle, representing an important area for future investigation. The effects of activin A on bone, however, remain controversial. We observed that reducing activin A expression in periodontal tissues prevented skeletal muscle atrophy but did not significantly affect bone loss. Although blocking activin A with the ActRIIA.muFc has been reported to promote bone formation[33,34], findings from our previous studies and others indicate a more complex, time-dependent role for activin A in osteogenesis[35,36]. Conflicting evidence also exists regarding its role in osteoclastogenesis[29,37]. Collectively, these findings highlight the context-dependent and multifaceted roles of activin A in bone remodeling, underscoring the need for further studies to clarify its effects on bone biology. In this study, we reduced activin A levels in periodontitis by locally silencing its expression via si*Inhba* injection. This localized approach may offer a safer therapeutic strategy compared to systemic administration of activin A-blocking antibodies, which may increase the likelihood of side effects. In fact, in a clinical trial involving patients with fibrodysplasia ossificans progressiva, deaths were reported following the administration of activin A-blocking antibodies[38]. Therefore, developing safer strategies to inhibit activin A is crucial, and local si*Inhba* injection could be a promising option.

RNA sequencing data analysis revealed seven genes that encode secretion-related proteins and are highly expressed in periodontitis. Besides *Inhba*, our primary focus in this study, these genes include *Mmp13*, *S100a8*, *S100a9*, *Spp1*, *Igf1*, and *Mctp1*. *Mmp13* is a key protease in periodontitis, known for its role in degrading the extracellular matrix, activating other matrix metalloproteinases (MMPs), and contributing to osteoclastogenesis[39]. *S100a8* and *S100a9*, primarily secreted by myeloid and epithelial cells, form a complex that triggers inflammation and exhibits antimicrobial effects[40]. *Spp1* (osteopontin) is secreted by osteoblasts and osteoclasts and facilitates osteoclast adhesion to bone surfaces[41]. Our scRNA-seq data analysis additionally revealed that *Spp1* is also expressed by fibroblasts and immune cells in inflamed gingiva, suggesting broader cellular sources of osteopontin in periodontitis. To our knowledge, elevated *Igf1* expression in periodontitis has not been previously reported, although IGF1 has been shown to enhance tissue regeneration in periodontal ligament cells[42]. Further research is needed to determine whether increased *Igf1* expression in periodontal tissue leads to systemic release and to elucidate its potential systemic effects. Finally, although *Mctp1* is known to regulate neurotransmitter release[43], our scRNA-seq dataset revealed relatively high *Mctp1* expression in endothelial cells, suggesting a previously unrecognized function in the periodontal microenvironment during inflammation.

In summary, our findings indicate that periodontitis induces changes in cell populations and their gene expression patterns within periodontal tissues, leading to increased secretion of activin A, which can then circulate through the bloodstream and potentially cause systemic muscle atrophy (Fig. 7). While it remains to be determined which substances secreted during local periodontitis contribute to systemic bone loss, identifying periodontitis as a potential cause of systemic muscle loss and confirming activin A as a key contributing factor, will be essential for developing effective treatments to prevent periodontitis-induced systemic muscle loss.

## Methods

### Human serum
Serum samples were obtained from individuals who visited Seoul National University Dental Hospital between January 1, 2021 and June 30, 2025. Periodontitis status and severity were determined based on clinical diagnosis and panoramic radiographs, with radiographic staging assigned according to the 2017 World Workshop on the Classification of Periodontal and Peri-Implant Diseases and Conditions[44]. Cohort demographics are summarized in Supplementary Table 3. Groups were age- and sex-matched. Exclusion criteria included self-reported history of malignancy, chronic infection, uncontrolled diabetes, and autoimmune disease.

### Korea national health and nutrition examination survey analysis
Cross-sectional data from the Korea National Health and Nutrition Examination Survey VII (2016-2018), conducted by the Korea Centers

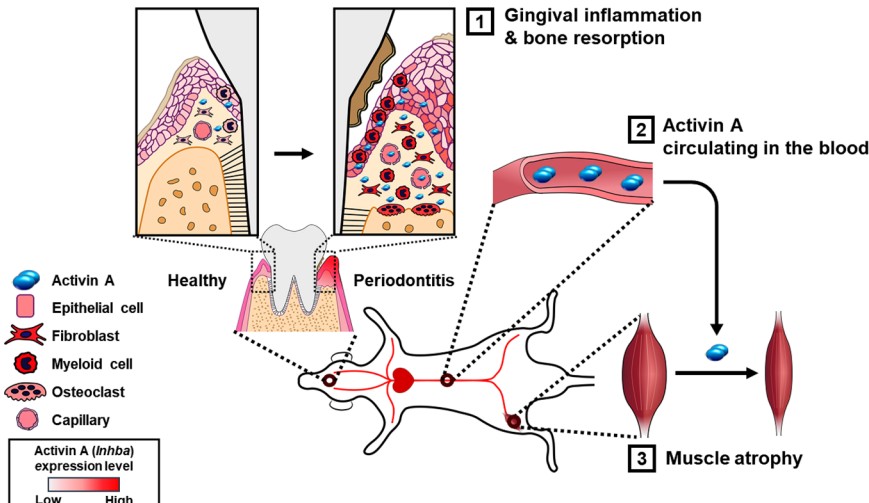

**Fig. 7 | Schematic diagram showing how activin A expression increases during the progression of periodontitis, leading to systemic muscle atrophy.** As periodontitis progresses, the epithelial cell layer proliferates, and myeloid cells infiltrate this region. In the connective tissue, the proliferation of fibroblasts and myeloid cells, along with an increased vascular distribution, is observed. Some myeloid cells specialize into osteoclasts, which resorb the alveolar bone. The expression levels of activin A in these cells are represented by color, with deeper red indicating higher expression levels. Although a small amount of activin A expression is observed in normal periodontal tissue, its expression in individual cells increases as periodontitis progresses. Simultaneously, the proliferation and recruitment of activin A-expressing cells occur, leading to a significant overall increase in activin A expression (step 1). Consequently, more activin A is secreted into the bloodstream (step 2), leading to systemic skeletal muscle atrophy (step 3).

for Disease Control and Prevention, were analyzed[45]. Detailed demographic characteristics are provided in Supplementary Table 2. Survey sampling weights were applied, and the complex survey design was accounted for in variance estimation to ensure national representativeness. Statistical analyses were performed using SPSS version 29.0.1 (IBM) with the complex-sample generalized linear model framework. Handgrip strength was defined as the maximum value recorded across measurements. To evaluate the association between periodontitis and handgrip strength, models were adjusted for age, sex, body mass index, education level, income, diagnosed diabetes, lifetime smoking, and frequency of muscle-strengthening exercise.

### Ligature-induced periodontitis

Periodontitis was induced in 8-week-old male C57BL/6 mice by ligating 5-0 black silk (Covidien) around the cervical region of the maxillary second molar, as previously described[10]. Mice were anesthetized with ketamine (100 mg/kg) and xylazine (10 mg/kg) prior to ligature placement. For intra-gingival siRNA injection, injection solution containing 5 μg siRNA and 0.8 μL in vivo-jetPEI (Polyplus) in 20 μl 5% glucose solution was administered into the palatal and buccal gingiva on both sides of the maxilla. Injections were performed three times at three-day intervals, starting on the day of ligature placement. Food intake was measured by providing each mouse with 150 g of standard chow and weighing the remaining food weekly to calculate weekly food consumption.

### Postprandial blood test

Mice were fasted overnight with water ad libitum and then allowed to feed for 30 minutes before blood collection by cardiac puncture. Plasma was collected in lithium-heparin tubes (BD) and analyzed for total protein, total cholesterol, triglycerides, glucose, and albumin using an AU480 Chemistry Analyzer (Beckman Colter).

### Grip strength analysis

Grip strength was measured using a Grip Strength Meter (Bioseb). Mice grasped a metal grid with all limbs and were gently pulled horizontally. Five consecutive measurements were recorded per session, and the maximum value was retained. Three sessions were performed per mouse, and the mean of the three maxima was used for analysis.

### AAV transduction

The transfer plasmids (pAAV-EF1a-*EGFP*, pAAV-EF1a-*Inhba*, pAAV-EF1a-*Inhba-EGFP*, and pAAV-EF1a-*Inhba-FLAG*) were constructed by cloning the respective cDNA sequences into the pAAV-EF1a backbone. For *Inhba*, the full-length coding sequence was used as the insert in all experiments. To verify sequence integrity, all constructs were confirmed by Sanger sequencing using the primers Ef1a_F (5'-TCA GCC TCA GA CAG TGG TTC-3') and WPRE_R (5'-CAT AGC GTA AAA GGA GCA ACA-3').

Recombinant AAV vectors were produced in AAV293T cells using a standard calcium phosphate transfection method. Cells were transfected with the AAV rep-cap plasmid (pRC-DJ, gift from M. Kay), the AAV helper plasmid (pHELPER, Agilent), and the respective transfer plasmid at a 1:1:1 molar ratio. At 72 hours post-transfection, viral particles were harvested from both the cell lysate and the culture medium. Purification was performed using a modified aqueous two-phase partitioning method[46]. The purified virus was buffer-exchanged and concentrated in PBS using centrifugal filters (Millipore).

Samples were treated with DNase I followed by thermal lysis. Quantitative PCR was performed using SYBR Green qPCR Master Mix (GLPBIO) with the following primers: WPRE-F (5'-GTG GAT ACG CTG CTT TAA TGC C-3') and WPRE-R (5'-CTC CTC ATA AAG AGA CAG CAA CCA G-3'). A standard curve was generated using serial dilutions of a linearized plasmid standard.

For gingival transduction, a total of $2 \times 10^{10}$ genome copies in 20 μL PBS per mouse were administered into both the palatal and buccal gingiva on each side of the maxilla.

### DXA and micro-CT analysis

DXA scanning was performed on days 0, 7, and 14 using an InAlyzer scanner (Medikors) with manufacturer-provided InAlyzer software for analysis. Micro-CT scans were conducted using Skyscan 1272 (Bruker-MicroCT) at a resolution of 13.3 μm (80 kV, 125 μA, 1 mm aluminum filter) for alveolar bone, and 6 μm (70 kV, 142 μA, 0.5 mm aluminum filter) for femurs and vertebrae. Alveolar bone resorption was assessed by measuring the distance between the CEJ and the ABC, using the

palatal groove of the ligature-tied second molar as a reference point. Measurements were performed using ImageJ software. Trabecular and cortical analyses of the femur were conducted in regions 0.5–2.5 mm and 1.5–2.5 mm from the primary spongiosa, respectively, while vertebral trabecular analysis was confined to the vertebral body excluding the growth plate. ROI selection and histomorphometric analysis were conducted using CTAn (Bruker-MicroCT), and images were generated with CTVox (Bruker-MicroCT).

## RNA fluorescence in situ hybridization (RNA FISH)
Three days after ligature placement, maxillae were fixed in 4% PFA, decalcified in 10% EDTA (pH 7.4) for 2 weeks, cryoprotected in 30% sucrose, embedded in OCT, and sectioned at 20 μm. Sections were permeabilized with 0.1% Triton X-100 for 5 minutes and hybridized overnight at 42 °C with DIG-labeled *Inhba* RNA probe in hybridization solution (50% formamide, 5X SSC, 5X Denhardt's solution, 0.5 mg/mL Herring Sperm DNA, 0.25 mg/mL E. coli tRNA). Post-hybridization washes were performed with SSC buffers at 62 °C, followed by washes in MABT (maleic acid buffer with 0.1% Tween 20) and blocking in RNA FISH blocking solution (5% normal goat serum, 0.1% Tween 20 in maleic acid buffer) for 30 minutes at room temperature (RT). Sections were incubated overnight at 4 °C with anti-DIG antibody (1:500, Roche, 11333089001), followed by incubation with donkey anti-sheep Alexa 488 secondary antibody (1:200, Invitrogen, A-11015) for 1 h at RT. Subsequently, sections were incubated overnight at 4 °C with anti-Ki-67 antibody (1:1,000, Abcam, ab15580) diluted in Ki-67 IF blocking solution (5% normal goat serum, 0.1% Triton X-100 in PBS), followed by goat anti-rabbit Alexa 594 secondary antibody (1:400, Invitrogen, A-11012) for 1 h at RT. Sections were stained with DAPI (Sigma-Aldrich), and signals were visualized using Zeiss Lattice SIM 5 microscope. The probe sequence for *Inhba* FISH is provided in Supplementary Data 1.

## Immunofluorescence (IF) staining
Samples for CD11b IF staining were prepared using the same protocol described for *Inhba* RNA FISH and Ki-67 IF staining, including tissue fixation, decalcification, and sectioning. Sections were permeabilized and blocked with CD11b IF blocking solution (10% normal goat serum, 0.1% Triton X-100 in PBS) for 30 minutes at RT. Sections were incubated overnight at 4 °C with anti-CD11b antibody (1:100, Invitrogen, 14-0112-82). After PBS washes, sections were incubated with goat anti-rat Alexa 488 secondary antibody (1:500, Invitrogen, A-11007) for 1 hour at RT and stained with DAPI. For muscle staining, tissues were snap-frozen in liquid nitrogen-cooled isopentane, sectioned, and fixed. Sections were permeabilized, blocked, and incubated overnight at 4 °C with anti-FLAG antibody (1:400, Cell Signaling Technology, D6W5B), followed by goat anti-rabbit Alexa 594 secondary antibody (1:500, Invitrogen, A-11012). For colocalization with activin receptors, muscle sections were incubated with anti-FLAG together with anti-ACVR2A (1:15, R&D Systems, AF340) and anti-ACVR2B (1:15, R&D Systems, AF339) antibodies, followed by goat anti-rabbit Alexa 594 secondary antibody (1:500, Invitrogen, A-11012) and donkey anti-goat Alexa Plus 488 secondary antibody (1:500, Invitrogen, A-32814). GFP was stained by incubating overnight at 4 °C with anti-GFP antibody (1:1,250, Cell Signaling Technology, 2555), followed by goat anti-rabbit Alexa 488 secondary antibody (1:500, Invitrogen, A-11008). Imaging was performed using Zeiss LSM 800 confocal microscope with Airyscan mode.

## Muscle weight and histology analysis
Pectoralis, triceps, quadriceps, and gastrocnemius muscles from both sides were dissected, and the average weight of each muscle was used for analysis. For muscle fiber diameter analysis, cross-sections were stained with hematoxylin and eosin, and the minimum Feret diameter was measured using ImageJ, with 250 fibers analyzed per region across three regions per muscle.

## Muscle fiber type analysis
Muscle cryosections were permeabilized, blocked, and incubated overnight at 4 °C with primary antibodies against MHC type I (1:100, DSHB, BA-D5) or a combination of MHC type IIa (1:100, DSHB, SC-71) and MHC type IIb (1:50, DSHB, BF-F3). Sections were then incubated with appropriate secondary antibodies. For type I staining, goat anti-mouse IgG(H + L) Alexa Fluor 594 (1:200, Invitrogen, A-11032) was used. For type IIa/IIb staining, goat anti-mouse IgG(H + L) Alexa Fluor 594 and goat anti-mouse IgM Alexa Fluor 488 (1:200, Invitrogen, A-21042) were used.

## Three-point bending test
Femoral biomechanical properties were assessed by three-point bending using an Instron E10000 testing system. Excised femurs were positioned with the anterior surface facing downward on two lower supports spaced 6 mm apart. A monotonic load was applied at the mid-diaphysis at a constant displacement rate of 2 mm/min until fracture. Maximum load and stiffness were determined from the force-displacement curve.

## Serum ELISA
Mouse blood samples were collected via cardiac puncture. Serum was obtained by centrifuging the blood samples for 10 min at 5,000 rpm in BD Microtainer SST Tubes (BD). The serum was snap-frozen in liquid nitrogen and stored at −80 °C. Mouse and human serum Activin A levels were measured using Quantikine Activin A ELISA Kit (R&D Systems) following the manufacturer's instructions.

## Luciferase assay
C3H10T1/2 cells were co-transfected with pGL3-(CAGA)$_{12}$-luc and pLEXm plasmids using Lipofectamine 3000 (Invitrogen)[47]. For activin A stimulation, recombinant activin A (R&D Systems) was added at 20 h post-transfection. Luciferase activity was measured at 24 h using the Luciferase Assay Reagent (Promega) according to the manufacturer's protocol.

## Osteoclast culture
Bone marrow cells were isolated as previously described[48,49]. Briefly, non-adherent cells were cultured in growth medium (α-MEM containing 10% FBS and 1% Penicillin Streptomycin) supplemented with M-CSF (25 ng/mL, PeproTech) for 3 days to generate bone marrow-derived macrophages (BMMs). Adherent BMMs were harvested and plated for osteoclast differentiation in growth medium supplemented with M-CSF (50 ng/mL) and RANKL (50 ng/mL, PeproTech).

## Single-cell RNA-sequencing
At day 3 after ligature placement, mice were euthanized and blood was drained by cardiac puncture to minimize blood contamination. Palatal gingiva was dissected, minced into small pieces (~0.5 mm²), and incubated in 0.25% trypsin-EDTA (Gibco) at 37 °C for 10 min. Tissues were washed and digested in DMEM containing dispase II (1.2 U/mL, Gibco) and collagenase type IV (2 mg/mL, Gibco) at 37 °C for 30 min. The digested tissue was triturated by gentle pipetting to generate a single-cell suspension, pelleted by centrifugation, and resuspended in PBS containing 1% BSA. Cell suspensions were filtered through a 40-μm cell strainer (Bel-Art), centrifuged, and resuspended in PBS with 0.1% BSA. Viable cells were counted by trypan blue exclusion, and cells were loaded targeting for sequencing 15,000 cells per sample. Single-cell libraries were prepared using the Chromium GEM-X Single Cell 3' v4 Gene Expression kit (10x Genomics) and sequenced according to the manufacturer's protocol.

## RNA sequencing dataset analysis
The scRNA-seq data (GSE315263) generated in this study were processed using the 10x Genomics Cell Ranger pipeline (v9.0.1) and

aligned to the reference genome. Previously published RNA sequencing datasets were downloaded from NCBI: GSE186882[15] (mouse bulk RNA-seq), GSE223924[19] (human bulk RNA-seq), GSE152042[20] (human scRNA-seq), GSE164241[21] (human scRNA-seq). For GSE152042, data from healthy and severe periodontitis patients were selected. Sequenced reads from GSE186882 and GSE152042 were aligned to reference genomes using the STAR aligner[50], while author-provided aligned read counts were used for GSE223924 and GSE164241. For bulk RNA sequencing data analysis, differential gene expression analysis was performed using edgeR[51] with quasi-likelihood F-tests. In the mouse bulk RNA-seq dataset, the top 25% of genes by expression were selected based on FPKM values. The GO term 'Secretion' (GO:0046903) from Mouse Genome Informatics was used to identify secretion-related proteins. Aligned reads from single-cell RNA sequencing data were analyzed using the Seurat package[52] (version 5) in R. Integration was conducted with the anchor-based RPCA method, and sketch-based analysis was applied to handle large human datasets. Cell clusters were classified using author-provided cell markers in human datasets. For cell type-specific differential expression analysis, pseudobulking was performed with the AggregateExpression function, followed by differential gene expression analysis with DESeq2[53], using the Wald test with Benjamini-Hochberg correction for multiple testing. Demographics of human subjects are provided in Supplementary Tables 4 and 5.

### Quantitative reverse transcription polymerase chain reaction (qRT-PCR)
Tissues were homogenized in QIAzol Lysis Reagent (Qiagen), and cultured cells were lysed directly in QIAzol. Total RNA was extracted using AccuPrep (Bioneer) and reverse transcribed into cDNA using the PrimeScript RT Reagent Kit (Takara). qRT-PCR was performed on QuantStudio 3 Real-Time PCR systems (Applied Biosystems) using TB Green Premix Ex Taq II (Takara). Relative gene expression was calculated using the $2^{(-\Delta\Delta Ct)}$ method and normalized to 18S rRNA expression. Primers are listed in Supplementary Table 6.

### Immunoblotting
Tissues were homogenized in Pro-Prep Protein Extraction Solution (Intron Biotechnology) supplemented with PhosSTOP phosphatase inhibitor cocktail (Roche). Equal amounts of protein were separated by SDS-PAGE, transferred to PVDF membranes, and incubated with primary antibodies overnight at 4 °C, followed by HRP-conjugated secondary antibodies. Signals were detected using SuperSignal chemiluminescent reagents (Thermo Scientific). Primary antibodies were anti-INHBA (1:1,000, Abcam, ab128958), anti-phospho-SMAD3 (1:1,000, Cell Signaling Technology, 9520), anti-MuRF1 (1:1,000, Santa Cruz, sc-32920), and anti-β-actin(C4)-HRP (1:1,000, Santa Cruz, sc-47778 HRP). The secondary antibody was goat anti-rabbit IgG (H + L)-HRP (1:4,000, Invitrogen, 31460).

### Statistics and reproducibility
Mean ± standard error of the mean (SEM) was presented for all values. Groups of three or more were analyzed using one-way ANOVA, while groups of two were analyzed using a Student's t-test. $P < 0.05$ was considered significant. At least three replicates were performed for all experiments. All attempts at replication were successful.

### Ethics
The study protocol for human serum analysis was reviewed by the Institutional Review Board of Seoul National University and determined to be exempt (IRB No. ERI25037) as it involved retrospective analysis of de-identified biobanked specimens and associated clinical data. All participants had provided written informed consent for biobanking at the time of sample collection. The study protocol for KNHANES analysis was reviewed by the Institutional Review Board of Seoul National University and determined to be exempt (IRB No. ERI26005) as it involved secondary analysis of publicly available de-identified data. All animal studies were approved by the Institutional Animal Care and Use Committee of Seoul National University (SNU-220914-4-1, SNU-230410-4-5, SNU-260102-1).

### Reporting summary
Further information on research design is available in the Nature Portfolio Reporting Summary linked to this article.

## Data availability
The single-cell RNA-seq dataset generated in this study has been deposited in the GEO database under accession number GSE315263. Previously generated RNA sequencing datasets are available in the GEO database under the accession codes: GSE152042, GSE164241, GSE186882, GSE223924. The remaining data are available within the article, Supplementary information or Source data file. Full uncropped scans of any cropped blot images are provided in the Supplementary information or Source data file. Source data are provided with this paper.

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

## Acknowledgements

We thank Seung Hyun Han (Seoul National University) and Hong-Hee Kim (Seoul National University) for assistance in setting up the ligature-induced periodontitis mouse model; Dongryeol Ryu (Gwangju Institute of Science and Technology), Sun Young Kim (Seoul National University Dental Hospital), and Kwang-Pyo Lee (Korea Research Institute of Bioscience and Biotechnology) for helpful discussions; Kyung-Hwan Kong (Seoul National University) for assistance with mouse husbandry; and Yong Jae Kim, Gi Su Eom, Hyoung Seuk Park, and Hwa Mok Paik (Zeiss) for assistance with SIM imaging. The biospecimens and data used

for this study were provided by the Biobank of Seoul National University Dental Hospital, a member of the Korea Biobank Network (project No. 2024ER050701). Y.-S.L. discloses support for the research of this work from the National Research Foundation of Korea (NRF) [grant numbers RS-2024-00336924 and RS-2025-02214577], funded by the Korean government (Ministry of Science and ICT).

## Author contributions

Y.-S.L. conceived and designed the study; W.S. performed most of the experiments with assistance from J.S., H.K.K. and N.-K.K.; W.S., Y.K., Y.T.J., M.-S.K., S.-J.L. and Y.-S.L. analyzed data; and W.S., J.S. and Y.-S.L. wrote the paper.

## Competing interests

The authors declare no competing interests.
