## [Transparent Peer Review file · Nature Communications]

Periodontitis induces skeletal muscle atrophy by increasing circulating levels of activin A

Corresponding Author: Professor Yun-Sil Lee

Version 0:

Reviewer comments:

Reviewer #1

(Remarks to the Author)

This manuscript showed that periodontitis significantly reduces muscle and bone mass without affecting fat mass or food intake using ligature-induced periodontitis mouse model. The authors demonstrated that activin A might play an important role for the development of muscle atrophy in this model and local injection of siRNA to reduce activin A expression can prevent the development of muscle atrophy. Authors also demonstrated that activin A expression was significantly increased in the gingiva of human individuals with periodontitis, compared to that in healthy human individuals. This manuscript is interesting and important for developing new therapeutic target to prevent sarcopenia induced by periodontitis. However, we have several concerns.

1) Expression of active A is significantly increased in mice treated with Porphyromonas gingivalis LPS (PG-LPS)? PG-LPS treated mice model is also a widely used mice model of periodontitis (Kristine Y et al. *Physiol Rep* 1(5), e00079, 2013) and shows muscle atrophy (Kawamura N et al. *J Physiol Sci* 69(3), 503-511, 2019).

2) The weight of triceps, quadriceps, and gastrocnemius were slightly but significantly reduced in ligature-induced periodontitis group as shown in Fig 2a. In order to confirm the fidelity of the data, the authors should show the data of cross sectional area.

3) Ligature-induced periodontitis altered the composition of myosin heavy chain isoform (I, IIa, IId/x, IIb) in skeletal muscle?

4) Increased expression of activin A in the gingiva of periodontitis was shown at mRNA level (qPCR). The authors should confirm this data at protein level (western blotting). In addition, the authors should confirm the local silencing of intra-gingival injection of siRNA by western blotting.

5) The authors should perform the mechanistic study about the skeletal muscle atrophy induced by activin A in periodontitis.

6) The authors examined the loss of skeletal muscle weight in pectoralis, triceps, quadriceps and gastrocnemius in ligature-induced periodontitis group. However, the magnitude of the decrease was different among the skeletal muscles. The authors should discuss about the different magnitude of the muscle atrophy among the skeletal muscles.

7) The decrease of skeletal muscle weight is specific for fast-twitch skeletal muscle specific or slow-twitch skeletal muscle ?

8) How did the authors determine the experimental condition of intra-gingival siRNA injection (Page 17)?

Reviewer #2

(Remarks to the Author)

In the manuscript "Periodontitis induces skeletal muscle atrophy by increasing circulating levels of activin A," the authors investigate the mechanisms by which periodontitis leads to systemic muscle atrophy, with a particular focus on the role of activin A.

Using a mouse model of periodontitis (LIP), the authors confirmed that periodontitis results in the loss of muscle and bone mass, as previously reported, and found that activin A—known to cause muscle atrophy—is highly expressed in the periodontitis-affected gingiva in this model. Furthermore, they demonstrated that suppressing activin A expression in periodontal tissue prevents muscle loss. These findings suggest that activin A plays a key role in periodontitis-induced muscle atrophy and that local siRNA administration to periodontal tissue may offer a therapeutic strategy to prevent this condition without causing systemic side effects.

As noted above, this study is commendable for providing scRNA-seq data on periodontitis-associated muscle atrophy and proposing a novel therapeutic approach. However, several points should be addressed to strengthen the conclusions:

1. While the authors argue that reduced food intake is not responsible for the muscle atrophy observed in periodontitis, differences in masticatory efficiency may lead to variations in nutrient absorption, even if total food consumption remains similar. To definitively exclude differences in nutrient uptake, fecal analysis or postprandial blood nutrient measurements would be valuable. These data would reinforce the conclusion that nutrient absorption is not affected.

Although Activin A has been implicated in muscle atrophy in several experimental models, as the authors mention, it remains uncertain whether the same mechanism applies directly to the periodontitis model used here. Additional evidence is needed to support the idea that Activin A-mediated muscle atrophy occurs in this specific context. If Activin A produced from inflamed gingiva indeed acts remotely on skeletal muscle via elevated circulating levels, the following assessments should be performed in both the periodontitis model and siRNA-treated mice:

2. Evidence of increased Activin signaling in muscle tissue, such as Smad phosphorylation or altered expression of Activin-responsive genes.
3. Surrogate markers indicating elevated systemic Activin levels, such as changes in serum FSH levels or hemoglobin concentrations.

Minor points:

4. While muscle atrophy was assessed by measuring four muscle groups, it would be informative to report whether similar changes were observed in other tissues, such as the heart, liver, kidney, or other non-muscle organs.
5. Please clarify the source of the scRNA-seq data used for gingival tissue analysis in the mouse periodontitis model (Line 140). Was this dataset generated from the same mice in which muscle atrophy was confirmed? If the data are from a public repository, please indicate this clearly and provide the appropriate accession number (e.g., GSE number).
6. Additional methodological details regarding the RNA-seq analysis of gingival tissue should be included in the Methods section. For the GO term "Secretion," please provide the corresponding GO identifier to improve clarity and reproducibility.

Reviewer #3

(Remarks to the Author)

Reviewer #4

(Remarks to the Author)

The study by Wonn Shim et al. employs a mouse ligature-induced periodontitis (LIP) model to investigate the role of *Inhba* upregulation in periodontal tissues and its systemic effects on muscle and bone atrophy. While the findings provide some insights into potential periodontal-muscle crosstalk, several critical issues need to be addressed to substantiate the claims and enhance the study's impact.

Major Concerns:

1. A major limitation is the exclusive reliance on mouse models without human validation. The translational relevance of these findings remains uncertain given known interspecies differences in periodontal pathophysiology and systemic responses. Clinical cohort studies demonstrating similar muscle/bone alterations and serum Activin A elevation in human periodontitis patients would significantly strengthen the biological relevance of these observations.
2. The study's novelty is constrained by its focus on Activin A, a well-established mediator of muscle atrophy. While the identification of periodontal tissues as a potential source adds some mechanistic insight, this finding alone does not substantially advance the field.
3. The proposed periodontal-muscle crosstalk lacks direct experimental evidence. The current data show *Inhba* upregulation in periodontal tissues but fail to demonstrate that periodontal-derived Activin A actually reaches and affects distant musculoskeletal tissues. Methodological approaches such as fluorescent tracer studies would be necessary to establish this critical link.
4. Potential confounding factors have not been adequately addressed. Periodontitis is known to elevate various SASP factors including S100A8 and MMP13, which could independently contribute to muscle senescence and bone loss. The study design does not include appropriate controls to isolate the specific contribution of Activin A from these other inflammatory mediators.

Minor Concerns:

1. Several technical issues require clarification. The muscle and bone phenotype characterization would be more convincing with functional assessments beyond simple mass measurements.
2. There is a discrepancy between the text description and Fig 1f regarding fat mass changes.
3. The single-cell RNA sequencing analysis lacks essential methodological details including sample size justification and demographic information, raising concerns about result stability and generalizability. Similarly, the RNA-seq data would

benefit from more comprehensive sample source documentation.

Version 1:

Reviewer comments:

Reviewer #1

(Remarks to the Author)

The authors sincerely responded to my comments and the manuscript was greatly improved. I have no further criticisms.

Reviewer #2

(Remarks to the Author)

The authors have adequately addressed the concerns raised during the initial review and have substantially improved the manuscript through the addition of new data and clarifications. In particular, the revised version provides further mechanistic support for the involvement of activin signaling in skeletal muscle, including evidence of SMAD3 phosphorylation and the expression of atrophy-related genes. The additional experiments examining the systemic effects of gingiva-derived activin A and the expanded analyses of muscle phenotypes also strengthen the overall conclusions.

The authors have also clarified several methodological issues, including the description of the datasets used for single-cell RNA-seq analysis and the experimental procedures for transcriptomic analyses. These revisions improve the transparency and reproducibility of the study.

Overall, the authors have responded appropriately to the reviewers' comments, and the manuscript has been significantly strengthened. In my opinion, the revised manuscript is suitable for publication in its current form.

Reviewer #3

(Remarks to the Author)

Reviewer #4

(Remarks to the Author)

The authors have done an exceptional job addressing all concerns raised in the previous round of review.

Key strengths of the revised manuscript:

1. Comprehensive experimental revisions: The authors have added functional assessments (grip strength, bone biomechanics), protein-level validation, and demonstrated activation of downstream Smad signaling in muscle tissue.
2. New high-quality scRNA-seq data: The replacement of public datasets with newly generated, high-resolution data from mice with confirmed muscle atrophy substantially strengthens the cellular localization findings.
3. Human validation: The addition of serum activin A measurements in periodontitis patients and epidemiological data linking periodontitis to reduced grip strength significantly enhances translational relevance.
4. Direct mechanistic evidence: The elegant FLAG-tagging experiment provides compelling direct evidence that gingiva-derived activin A reaches and signals in distant skeletal muscle, establishing the proposed periodontal-muscle crosstalk on solid experimental footing.

The authors have produced a rigorous, well-controlled, and conceptually novel study that establishes periodontitis as a previously unrecognized source of systemic activin A driving muscle atrophy.

Reviewer #1:

1. Expression of active A is significantly increased in mice treated with *Porphyromonas gingivalis* LPS (PG-LPS)? PG-LPS treated mice model is also a widely used mice model of periodontitis (Kristine Y et al. *Physiol Rep* 1(5), e00079, 2013) and shows muscle atrophy (Kawamura N et al. *J Physiol Sci* 69(3), 503-511, 2019).

Response: We appreciate the reviewer's insightful suggestion. To address this point, we administered PG-LPS via intraperitoneal (IP) injection at the same dose used by Kawamura et al. (0.8 mg/kg/day) and measured serum activin A levels by ELISA at 1, 3, and 7 days after injection, corresponding to time points at which we observed a significant increase in activin A in the ligature-induced periodontitis (LIP) model ($n = 6-8$). Under these experimental conditions, however, we did not detect a significant increase in circulating activin A levels in PG-LPS-treated mice compared with PBS-injected controls. **Text Redacted**

Importantly, in the PG-LPS-induced periodontitis model, PG-LPS is typically administered chronically for approximately one month, and Kawamura et al. reported a significant increase in phosphorylated Smad2/3 in the masseter muscle under these conditions. Thus, although acute PG-LPS exposure did not elevate circulating activin A in our short-term analysis, we cannot exclude the possibility that modest but sustained increases in activin A during prolonged PG-LPS exposure may contribute to muscle loss over the long term.

Nevertheless, we would like to emphasize that the primary scope of the present study is to investigate how local periodontitis drives systemic muscle atrophy, as modeled by ligature-induced periodontitis. In contrast, the PG-LPS model induces systemic inflammation directly through IP administration of PG-LPS, representing a distinct biological context. For this reason, incorporating the PG-LPS model could detract from the conceptual focus of the current manuscript. A detailed analysis of the PG-LPS model would therefore require a distinct experimental framework and fall beyond the primary scope of the present study. That said, we agree that this is an important and interesting direction, and we plan to pursue this model in a future, dedicated study.

2. The weight of triceps, quadriceps, and gastrocnemius were slightly but significantly reduced in ligature-induced periodontitis group as shown in Fig 2a. In order to confirm the fidelity of the data, the authors should show the data of cross sectional area.

Response: We thank the reviewer for this valuable suggestion, which allowed us to further strengthen the robustness of our findings. In the original manuscript, muscle cross-sectional area (CSA) analysis was performed only for the quadriceps (Fig. 2b-c), as this muscle exhibited the largest reduction in weight. In response to the reviewer's comment, we have now extended this histological CSA analysis to all four muscle groups examined in this study: the triceps, gastrocnemius, pectoralis, and quadriceps. Consistent with the muscle weight data, CSA analyses revealed reductions in muscle fiber size across all muscles in LIP mice compared with controls (Supplementary Fig. 2b-g), confirming that the observed decreases in muscle weight reflect true muscle atrophy. Furthermore, to determine whether these structural changes translate into functional impairment, we performed grip strength measurements. LIP mice exhibited significantly reduced grip strength compared with control mice (Fig. 2d), demonstrating that the reductions in muscle mass and fiber size are accompanied by a decline in muscle function.

3. Ligature-induced periodontitis altered the composition of myosin heavy chain isoform (I, IIa, IIc/x, IIb) in skeletal muscle?

Response: We thank the reviewer for this insightful suggestion, which enabled a more comprehensive characterization of periodontitis-induced skeletal muscle changes. To address this point, we performed muscle fiber type-specific immunostaining in the gastrocnemius muscle two weeks after ligature placement. The proportion of type I fibers was not altered in periodontitis-induced mice. Within the type II fiber population, we observed a modest reduction in type IIb fibers, accompanied by trends toward increased proportions of type IIa and type IIc fibers, suggesting a potential shift from predominantly glycolytic toward more oxidative fiber types (Supplementary Fig. 2i, k). However, because these changes in fiber type composition did not reach statistical significance, we interpret these results cautiously as indicative trends that warrant further investigation.

4. Increased expression of activin A in the gingiva of periodontitis was shown at mRNA level (qPCR). The authors should confirm this data at protein level (western blotting). In addition, the authors should confirm the local silencing of intra-gingival injection of siRNA by western blotting.

Response: We thank the reviewer for this valuable suggestion to validate our findings at the protein level. As recommended, we performed Western blot analyses of INHBA protein expression in gingival tissues from the control, LIP, LIP-siCtrl, and LIP-si*Inhba* groups. Because activin A is a homodimer composed of two INHBA subunits, INHBA protein levels directly reflect activin A abundance. In the revised manuscript, we included data showing that INHBA protein levels were markedly increased in the gingiva of periodontitis-induced mice compared with controls, thereby validating our qRT-PCR results at the protein level (Supplementary Fig. 3a). Furthermore, intra-gingival injection of si*Inhba* efficiently reduced INHBA protein levels in the gingiva compared with siCtrl-injected mice (Supplementary Fig. 6a), demonstrating successful local silencing of activin A at the protein level.

5. The authors should perform the mechanistic study about the skeletal muscle atrophy induced by activin A in periodontitis.

Response: We appreciate the reviewer's valuable comment regarding the need to explore the mechanism underlying skeletal muscle atrophy induced by activin A in periodontitis. Previous studies have demonstrated that both activin A and myostatin bind to the same activin type II receptors (ActRIIA and ActRIIB), which subsequently recruit type I receptors such as ALK4 or ALK5. This receptor complex initiates intracellular signaling via phosphorylation of Smad2/3, ultimately leading to increased expression of FOXO3, Atrogin-1 (encoded by *Fbxo32*), and MuRF1 (encoded by *Trim63*)—key mediators of muscle atrophy (Goodman et al., *Mol Endocrinol*, 2013; Lokireddy et al., *Am J Physiol Cell Physiol*, 2011 & 2012; McFarlane et al., *J Cell Physiol*, 2006).

In the revised manuscript, we included data showing that LIP or gingival overexpression of activin A via AAV delivery induces downstream activin signaling in both the quadriceps and extensor digitorum longus (EDL) muscles, characterized by increased SMAD3 phosphorylation, upregulation of atrophy-related genes (*Foxo3*, *Fbxo32*, and *Trim63*), and increased MuRF1 protein levels (Fig 3e-g, Supplementary Fig. 3b-d, j-o).

6. The authors examined the loss of skeletal muscle weight in pectoralis, triceps, quadriceps and gastrocnemius in ligature-induced periodontitis group. However, the

magnitude of the decrease was different among the skeletal muscles. The authors should discuss about the different magnitude of the muscle atrophy among the skeletal muscles.

Response: We thank the reviewer for this important observation. The differences in the magnitude of muscle atrophy among skeletal muscle groups may be attributed, at least in part, to muscle-specific differences in the expression of activin type II receptors (ActRII). A previous study reported that the distribution of activin type IIB receptor (ActRIIB) varies among different muscle types, which may influence their susceptibility to atrophy-inducing signals (Mendias et al., *J Appl Physiol*, 2006). More recently, it has been suggested that the expression of *Acvr1b* (ALK4) and *Tgfbr1* (ALK5), type I receptors mediating activin A signaling, also differs across muscle groups. For example, a recent preprint reports lower expression levels of *Acvr1b* and *Tgfbr1* in the soleus compared with the gastrocnemius (Jaspers et al., *Research Square*, 2024), consistent with muscle-specific differences in sensitivity to activin A-mediated signaling.

In addition, differences in muscle fiber-type composition may contribute to the observed variability in atrophy severity. The quadriceps, which exhibited the most pronounced atrophy, is predominantly composed of fast-twitch type II fibers. When we assessed fiber-type-specific atrophy, fast-twitch type II fibers were more susceptible to atrophy than slow-twitch type I fibers, with the greatest reduction in fiber diameter observed in type IIb (22.5%), followed by type IIx (19.7%), type IIa (13.2%), and type I fibers (7.0%) (Supplementary Fig. 2j). We have addressed this point in the Discussion section (line 426-436) of the revised manuscript.

7. The decrease of skeletal muscle weight is specific for fast-twitch skeletal muscle specific or slow-twitch skeletal muscle?

Response: We thank the reviewer for this insightful question. In the revised manuscript, we included data showing that fast-twitch type II fibers exhibited greater susceptibility to atrophy than slow-twitch type I fibers, with type IIb fibers showing the largest reduction in diameter (22.5%), followed by type IIx (19.7%) and type IIa fibers (13.2%), whereas type I fibers exhibited the smallest decrease (7.0%) (Supplementary Fig. 2j). This pattern is consistent with age-associated muscle atrophy, in which fast-twitch fibers are known to be more vulnerable than slow-twitch fibers (Lexell et al., *J Neurol Sci*, 1988). Although our study focused on periodontitis-induced muscle loss rather than aging per se, these similarities suggest that periodontitis may contribute to sarcopenia, a possibility that is particularly relevant given the high prevalence of periodontitis in older adults.

8. How did the authors determine the experimental condition of intra-gingival siRNA injection (Page 17)?

Response: The concentrations of siRNA and in vivo-jetPEI were determined according to the manufacturer's recommended protocol. The injection volume was empirically optimized to ensure efficient local delivery while minimizing diffusion to surrounding tissues. Specifically, pilot experiments using Evans blue dye were performed to assess distribution within the gingival tissue, and an injection volume of 5 μ L per site was selected, as it provided localized retention without detectable spread to adjacent tissues (Supplementary Fig. 3e).

Reviewer #2:

1. While the authors argue that reduced food intake is not responsible for the muscle atrophy observed in periodontitis, differences in masticatory efficiency may lead to variations in nutrient absorption, even if total food consumption remains similar. To definitively exclude differences in nutrient uptake, fecal analysis or postprandial blood nutrient measurements would be valuable. These data would reinforce the conclusion that nutrient absorption is not affected.

Response: We thank the reviewer for this insightful comment. As suggested, we performed postprandial blood nutrient analyses to evaluate whether nutrient absorption was altered in the ligature-induced periodontitis (LIP) model. Measurements of total protein, albumin, total cholesterol, and triglycerides revealed no significant differences between periodontitis-induced mice and controls (Supplementary Fig. 1a-d), indicating that nutrient absorption was not impaired. Interestingly, postprandial blood glucose levels increased as periodontitis progressed (Supplementary Fig. 1e). Given that protein and lipid parameters were unchanged, this finding further argues against impaired nutrient absorption due to reduced masticatory efficiency and may reflect a previously reported association between periodontitis and diabetes, in which periodontal inflammation negatively affects glycemic control (Preshaw et al., *Diabetologia*, 2011).

2. Evidence of increased Activin signaling in muscle tissue, such as Smad phosphorylation or altered expression of Activin-responsive genes.

Response: We thank the reviewer for this important comment. In the revised manuscript, we added Western blot analyses demonstrating that SMAD3 phosphorylation is significantly increased in skeletal muscle following periodontitis induction, providing direct evidence of enhanced activin signaling in muscle (Fig. 3e, Supplementary Fig. 3b). Intra-gingival injection of *siInhba* markedly reduced SMAD3 phosphorylation in skeletal muscle, confirming that this signaling activation is dependent on gingiva-derived activin A (Supplementary Fig. 6b). In addition, gingival overexpression of activin A via AAV delivery induced downstream activin signaling in both the quadriceps and EDL muscles, as evidenced by increased SMAD3 phosphorylation, upregulation of atrophy-related genes (*Foxo3*, *Fbxo32*, and *Trim63*), and elevated MuRF1 protein levels (Supplementary Fig. 3j-o).

3. Surrogate markers indicating elevated systemic Activin levels, such as changes in serum FSH levels or hemoglobin concentrations.

Response: We thank the reviewer for this thoughtful suggestion. All experiments in this study were performed in male mice. As suggested, we attempted to measure serum FSH levels as a potential surrogate marker of elevated systemic activin A using commercially available mouse FSH ELISA kits (including the Invitrogen Mouse FSH ELISA kit, #EEL097). However, serum FSH levels in male mice were below the detection limit of these assays, precluding reliable quantification. This result was not entirely unexpected, as regulation of circulating FSH by activin A has been more prominently described in females, whereas serum FSH levels in males are typically low and less responsive (Elsholz et al., *Hum Reprod*, 2004).

We also measured hemoglobin concentrations to assess whether systemic activin A elevation was associated with hematological changes. No significant differences in hemoglobin levels were observed between control and periodontitis-induced mice. Notably, increases in red blood cell parameters reported with Fc-tagged decoy activin

type II receptor (ActRII) treatments, including luspatercept and sotatercept, are thought to be mediated predominantly through blockade of ligands such as GDF11 rather than activin A itself (Dussiot et al., *Nat Med*, 2014; Suragani et al., *Nat Med*, 2014), highlighting the complexity of using hematological parameters as surrogate markers of systemic activin A activity.

At present, there is no well-established surrogate biomarker that reliably reflects elevations in circulating activin A independent of sex. Identifying such markers would require dedicated investigation beyond the scope of this study. Therefore, we focused on demonstrating downstream activation of activin signaling in skeletal muscle, including increased SMAD2/3 phosphorylation, which provides mechanistically relevant evidence of systemic activin A activity.

Minor points:

4. While muscle atrophy was assessed by measuring four muscle groups, it would be informative to report whether similar changes were observed in other tissues, such as the heart, liver, kidney, or other non-muscle organs.

Response: We thank the reviewer for this insightful comment. Activin A is known to influence multiple organs beyond skeletal muscle, including the heart, liver, and kidney. While the present study focused specifically on skeletal muscle, we agree that examining whether periodontitis-induced systemic activin A affects other non-muscle organs would provide additional insight. However, such analyses were beyond the scope of the current study. We have therefore addressed this important point in the Discussion section (line 453-457) and plan to investigate the effects of periodontitis-induced activin A on other organs in future studies.

5. Please clarify the source of the scRNA-seq data used for gingival tissue analysis in the mouse periodontitis model (Line 140). Was this dataset generated from the same mice in which muscle atrophy was confirmed? If the data are from a public repository, please indicate this clearly and provide the appropriate accession number (e.g., GSE number).

Response: We thank the reviewer for this important comment regarding the source of the mouse gingival scRNA-seq data. In the original manuscript, the gingival scRNA-seq data were obtained from a publicly available GEO dataset (GSE228635). Although the accession number was provided in the Methods and Data Availability sections, we agree that this should have been stated more explicitly in the Results section. These data were generated using a ligature-induced periodontitis (LIP) model similar to ours; however, muscle atrophy was not assessed in those mice.

To address this limitation, we have now generated new gingival scRNA-seq data from LIP mice in which muscle atrophy was confirmed (Fig. 4a-d, Supplementary Fig. 4) and have replaced the previously used public dataset (GSE228635) with this new dataset (GSE315263). Analysis of this newly generated dataset validated our original findings, showing significant upregulation of *Inhba* in fibroblasts, myeloid cells, and epithelial cells, particularly within proliferating epithelial cell populations, in periodontitis-affected gingiva. Importantly, the new dataset provides substantially improved resolution, with three biological replicates per group (total 88,191 cells) compared with single samples per group in the public dataset (total 3,514 cells). This enabled more robust assessment of biological variability and identification of additional cell populations, including a junctional epithelial cell population with particularly high *Inhba* expression.

The GEO accession number (GSE315263) for this newly generated dataset has been included in the Methods (line 686) and Data Availability sections (line 731-732) of the revised manuscript. Although the dataset has not yet been publicly released, it can be accessed using the GEO private token (klmnkmqwfrendkj) should the reviewer wish to access the data.

6. Additional methodological details regarding the RNA-seq analysis of gingival tissue should be included in the Methods section. For the GO term “Secretion,” please provide the corresponding GO identifier to improve clarity and reproducibility.

Response: As suggested by the reviewer, we have provided the GO identifier corresponding to the term “Secretion” (GO:0046903) and included additional methodological details regarding the RNA-seq analysis of gingival tissue in the Methods section (line 696-697) of the revised manuscript to improve clarity and reproducibility.

Reviewer #3:

Response: Thank you for letting us know. We fully support this co-review initiative for Early Career Researchers. We are grateful for the time and effort both reviewers devoted to evaluating our work, and we truly appreciate the thoughtful comments and constructive feedback.

Reviewer #4:

1. A major limitation is the exclusive reliance on mouse models without human validation. The translational relevance of these findings remains uncertain given known interspecies differences in periodontal pathophysiology and systemic responses. Clinical cohort studies demonstrating similar muscle/bone alterations and serum Activin A elevation in human periodontitis patients would significantly strengthen the biological relevance of these observations.

Response: We appreciate the reviewer’s important comment regarding the translational relevance of our study. To address this concern, we have now generated new human validation data. First, we measured serum activin A levels in healthy individuals and periodontitis patients with alveolar bone loss extending to the mid-third of the root and beyond (Fig. 4g) and found that circulating activin A levels were significantly elevated in the periodontitis group. This finding indicates that the systemic increase in activin A observed in our mouse model is also present in humans with periodontitis. In addition, analysis of data from the Korea National Health and Nutrition Examination Survey (KNHANES) revealed that older individuals with periodontitis exhibited significantly reduced grip strength compared with age-matched, periodontally healthy controls (Supplementary Table 1). This population-level association between periodontitis and impaired muscle function is consistent with our experimental findings in mice. Together, these newly added human serum and epidemiological data support the translational

relevance of our study by demonstrating that the link between periodontitis, elevated systemic activin A, and muscle weakness extends beyond the mouse model.

2. The study's novelty is constrained by its focus on Activin A, a well-established mediator of muscle atrophy. While the identification of periodontal tissues as a potential source adds some mechanistic insight, this finding alone does not substantially advance the field.

Response: We appreciate the reviewer's insightful comment and agree that activin A is a well-established mediator of muscle atrophy. However, the novelty of our study does not lie in redefining the role of activin A itself, but in identifying periodontitis, a highly prevalent chronic inflammatory condition, as a previously unrecognized source of systemic activin A elevation. To address concerns regarding novelty and translational relevance, we have strengthened the manuscript in several ways.

First, we directly demonstrated that gingiva-derived activin A can reach skeletal muscle and activate downstream signaling pathways using AAV-*Inhba* transduction in gingival tissues (Fig. 3i, Supplementary Fig. 3j-o). Second, intra-gingival injection of si*Inhba* markedly reduced SMAD3 phosphorylation in skeletal muscle (Supplementary Fig. 6b), confirming downregulation of activin A signaling in muscle. Finally, we added new human data demonstrating elevated serum activin A levels in patients with periodontitis (Fig. 4g), as well as an association between periodontitis and reduced grip strength in a population-based cohort (Supplementary Table 1), as described in our response to Major Concern 1.

Thus, rather than simply reaffirming the role of activin A in muscle atrophy, our study identifies periodontitis as a novel systemic source of activin A and integrates mechanistic, experimental, and human evidence linking periodontitis to muscle atrophy. We believe these findings substantially enhance both the conceptual novelty and the public health relevance of the work.

3. The proposed periodontal-muscle crosstalk lacks direct experimental evidence. The current data show *Inhba* upregulation in periodontal tissues but fail to demonstrate that periodontal-derived Activin A actually reaches and affects distant musculoskeletal tissues. Methodological approaches such as fluorescent tracer studies would be necessary to establish this critical link.

Response: We sincerely thank the reviewer for this valuable suggestion, which helped us strengthen the experimental support for the proposed periodontal-muscle crosstalk. To address this concern, we initially generated a pLEXm-*Inhba-EGFP* construct and assessed activin A signaling using a luciferase reporter assay following transfection with a Smad2/3-responsive pGL3-(CAGA)₁₂-luciferase expression construct. We found that EGFP-tagged activin A interfered with activin A-mediated downstream signaling (Supplementary Fig. 3f), indicating that EGFP tagging compromises biological activity. We therefore additionally generated a pLEXm-*Inhba-FLAG* construct, using a much smaller FLAG epitope to minimize potential steric interference. Although activin A-FLAG did not inhibit signaling, it also did not activate signaling to the same extent as untagged activin A. Based on these results, we used FLAG-tagged activin A exclusively for tracing purposes and employed untagged activin A to evaluate systemic muscle atrophy resulting from gingiva-derived activin A upregulation.

Next, we locally transduced gingival tissue with AAV-*Inhba-FLAG* to induce gingiva-specific expression of FLAG-tagged activin A. Immunofluorescence analysis

demonstrated that activin A-FLAG produced in gingival tissue was detectable in skeletal muscle, indicating that gingiva-derived activin A can reach distant muscle tissue. Notably, the FLAG signal colocalized with activin type II receptors at the muscle fiber membrane, suggesting receptor engagement of activin A-FLAG (Fig. 3i).

Finally, we delivered either AAV-*Inhba* or AAV-*EGFP* to gingival tissue. Mice receiving AAV-*Inhba* exhibited a significant reduction in muscle mass (Fig. 3j), accompanied by increased SMAD3 phosphorylation in skeletal muscle (Supplementary Fig. 3j,m), compared with mice receiving AAV-*EGFP*. These findings demonstrate that gingiva-derived activin A reaches skeletal muscle and induces muscle atrophy through activation of downstream signaling pathways.

We believe that this AAV-based, tissue-restricted tagging approach directly addresses the reviewer's concern by providing direct experimental evidence that activin A produced in periodontal tissue can reach and signal in distant muscle, thereby supporting the existence of activin A-mediated periodontal-muscle crosstalk.

4. Potential confounding factors have not been adequately addressed. Periodontitis is known to elevate various SASP factors including S100A8 and MMP13, which could independently contribute to muscle senescence and bone loss. The study design does not include appropriate controls to isolate the specific contribution of Activin A from these other inflammatory mediators.

Response: We appreciate the reviewer's important point regarding potential confounding factors. In the original manuscript, we showed that siRNA-mediated *Inhba* knockdown restored muscle mass to levels comparable to those of control mice (Fig. 6a-i). In the revised manuscript, we additionally confirmed that ligature-induced periodontitis does not induce skeletal muscle atrophy in transgenic mice overexpressing follistatin, an endogenous inhibitor of activin A (Fig. 6j-m). Together, these findings support a central role for activin A in mediating systemic muscle loss during periodontitis.

That said, we agree that other SASP factors, such as S100A8 and MMP13, could potentially contribute. To address this possibility, we measured serum levels of S100A8 and MMP13 using a mouse S100A8 ELISA kit (R&D Systems, #DY3059) and a mouse MMP13 ELISA kit (Abcam, #ab316904) in control mice, ligature-induced periodontitis (LIP) mice, and LIP mice injected with either siCtrl or si*Inhba* on day 7, a time point at which serum activin A levels were markedly elevated ($n = 8$). S100A8 levels were below the limit of detection in all groups. MMP13 levels (mean \pm SD, pg/mL) were 635.6 ± 104.7 in controls, 1025.4 ± 555.9 in the LIP group, 1117.0 ± 317.3 in the LIP-siCtrl group, and 853.5 ± 502.8 in the LIP-si*Inhba* group. Although MMP13 levels appeared higher in periodontitis-induced groups compared with controls, the substantial variability resulted in no statistically significant differences among groups. Notably, although *Mmp13* gene expression in periodontal tissue was robustly upregulated from day 1 to day 14 in the LIP model, exceeding the magnitude of *Inhba* induction (Fig. 3c), this increase was not reflected at the circulating protein level. This discrepancy suggests that the systemic effects of MMP13 are likely limited and less predictable than those of activin A. Consistent with its known biology, even when secreted, MMP13 is largely retained within the extracellular matrix and rapidly consumed during substrate degradation, resulting in minimal levels of free MMP13 in the circulation.

Taken together, these findings suggest that, in the LIP model, activin A exerts more prominent systemic effects than S100A8 or MMP13, and may therefore play a more substantial role in the muscle atrophy observed in this model.

Minor Concerns:

1. Several technical issues require clarification. The muscle and bone phenotype characterization would be more convincing with functional assessments beyond simple mass measurements.

Response: We thank the reviewer for this valuable suggestion. In the revised manuscript, we have added functional assessments to complement the measurements of muscle mass and bone micro-CT analyses. For muscle, we performed grip strength testing to evaluate muscle function (Fig. 2d, Fig. 6i, m). For bone, we conducted three-point bending tests to assess biomechanical strength (Fig. 2h, i, Supplementary Fig. 6m). Together, these additional functional data strengthen the characterization of both muscle and bone phenotypes.

2. There is a discrepancy between the text description and Fig 1f regarding fat mass changes.

Response: We appreciate the reviewer's comment regarding this point. We believe the current description is consistent with Fig. 1f, but we recognize that the wording, particularly the distinction between percentage fat mass (Fig. 1f) and absolute fat mass (Fig. 1g), may have caused confusion. We have revised the text (line 136-139) to clarify this distinction and to avoid any potential misunderstanding.

3. The single-cell RNA sequencing analysis lacks essential methodological details including sample size justification and demographic information, raising concerns about result stability and generalizability. Similarly, the RNA-seq data would benefit from more comprehensive sample source documentation.

Response: We appreciate the reviewer's comment regarding methodological transparency. For both human scRNA-seq and bulk RNA-seq analyses, we have included demographic information in the revised manuscript (Supplementary Table 4 and 5). For the human scRNA-seq, to address concerns regarding sample size and statistical robustness, we integrated all publicly available datasets generated using compatible platforms (10X Genomics and Illumina sequencing), resulting in a combined cohort of healthy ($n = 15$) and periodontitis ($n = 9$) samples. This integration increased sample size and improved statistical power. For the human bulk RNA-seq, we analyzed gingival tissue from 10 healthy controls and 10 individuals with periodontitis. Sample size was determined using RNASeqPower based on the negative binomial model described by Hart *et al.*, which indicated that ≥ 8 samples per group would be sufficient ($CV = 0.8$, $\alpha = 0.05$, power = 0.9). We have also provided more comprehensive documentation of sample sources, inclusion criteria, and data processing procedures in the Methods (line 685-705), Data Availability sections (line 733-734), and Reporting Summary of the revised manuscript.

For the mouse scRNA-seq analysis, we generated new datasets from our own samples with three biological replicates per group (Control vs. LIP), allowing assessment of biological variability (Fig. 4a-d, Supplementary Fig. 4). Importantly, this dataset provides

substantially improved resolution compared with previously analyzed public datasets, which consisted of a single sample per group and a total of 3,514 cells, whereas our dataset includes 88,191 cells. The GEO accession number for this newly generated dataset (GSE315263) has been included in the Data Availability sections (line 731-732) of the revised manuscript. Although the dataset has not yet been publicly released, it will be made publicly available immediately upon acceptance of the manuscript. In the meantime, should the reviewer wish to access the data prior to public release, it can be accessed using the GEO private token: klmnkmqwfrendkj.

References

- Dussiot, M., Maciel, T.T., Fricot, A., Chartier, C., Negre, O., Veiga, J., Grapton, D., Paubelle, E., Payen, E., Beuzard, Y., Leboulch, P., Ribeil, J.A., Arlet, J.-B., Coté, F., Courtois, G., Ginzburg, Y.Z., Daniel, T.O., Chopra, R., Sung, V., Hermine, O., & Moura, I.C. (2014). An activin receptor IIA ligand trap corrects ineffective erythropoiesis in β -thalassemia. *Nature Medicine*, 20(4), 398–407.
- Elsholz, D.D., Padmanabhan, V., Rosenfield, R.L., Olton, P.R., Phillips, D.J., & Foster, C. M. (2004). GnRH agonist stimulation of the pituitary–gonadal axis in children: age and sex differences in circulating inhibin-B and activin-A. *Human Reproduction*, 19(12), 2748–2758.
- Goodman, C.A., McNally, R.M., Hoffmann, F.M., & Hornberger, T.A. (2013). Smad3 induces Atrogin-1, inhibits mTOR and protein synthesis, and promotes muscle atrophy in vivo. *Molecular Endocrinology*, 27(11), 1946–1957.
- Hart, S.N., Therneau, T.M., Zhang, Y., Poland, G.A., & Kocher, J.P. (2013). Calculating sample size estimates for RNA sequencing data. *Journal of Computational Biology*, 20(12), 970–978.
- Jaspers, R., Shi, A., Hillege, M., Noort, W., Offringa, C., Wu, G., Forouzanfar, T., Hoogaars, W., & Wüst, R. (2024). Myofibre-specific knockout of TGF- β type I receptors triggers muscle hypertrophy and promotes contraction and oxidative metabolism. Preprint at *Research Square*. <https://www.researchsquare.com/article/rs-5136404/v1>
- Lexell, J., Taylor, C.C., Sjöström, M. (1988). What is the cause of the ageing atrophy? Total number, size and proportion of different fiber types studied in whole vastus lateralis muscle from 15- to 83-year-old men. *Journal of the Neurological Sciences*, 84(2-3), 275-94.
- Lokireddy, S., Mouly, V., Butler-Browne, G., Gluckman, P.D., Sharma, M., Kambadur, R., & McFarlane, C. (2011). Myostatin promotes the wasting of human myoblast cultures through promoting ubiquitin–proteasome pathway-mediated loss of sarcomeric proteins. *American Journal of Physiology - Cell Physiology*, 301(6), C1316–C1324.
- Lokireddy, S., Wijesoma, I.W., Sze, S.K., McFarlane, C., Kambadur, R., & Sharma, M. (2012). Identification of atrogin-1-targeted proteins during the myostatin-induced skeletal muscle wasting. *American Journal of Physiology - Cell Physiology*, 303(5), C512–C529.
- McFarlane, C., Plummer, E., Thomas, M., Henneby, A., Ashby, M., Ling, N., Smith, H., Sharma, M., & Kambadur, R. (2006). Myostatin induces cachexia by activating the ubiquitin proteolytic system through an NF- κ B-independent, FoxO1-dependent mechanism. *Journal of Cellular Physiology*, 209(2), 501–514.
- Mendias, C.L., Marcin, J.E., Calderon, D.R., & Faulkner, J.A. (2006). Contractile properties of EDL and soleus muscles of myostatin-deficient mice. *Journal of Applied Physiology*, 101(3), 898–905.
- Preshaw, P.M., Alba, A.L., Herrera, D., Jepsen, S., Konstantinidis, A., Makrilakis, K., & Taylor, R. (2011). Periodontitis and diabetes: a two-way relationship. *Diabetologia*, 55(1), 21–31.
- Suragani, R.N.V.S., Cadena, S.M., Cawley, S.M., Sako, D., Mitchell, D., Li, R., Davies, M.V., Alexander, M.J., Devine, M., Loveday, K.S., Underwood, K.W., Grinberg, A.V., Quisel, J.D., Chopra, R., Pearsall, R.S., Seehra, J., & Kumar, R. (2014). Transforming growth factor- β superfamily ligand trap ACE-536 corrects anemia by promoting late-stage erythropoiesis. *Nature Medicine*, 20(4), 408–414.

Reviewer #1:

The authors sincerely responded to my comments and the manuscript was greatly improved. I have no further criticisms.

- **Response:** We sincerely thank the reviewer for the positive evaluation and for the constructive comments, which have helped us strengthen our manuscript.

Reviewer #2:

The authors have adequately addressed the concerns raised during the initial review and have substantially improved the manuscript through the addition of new data and clarifications. In particular, the revised version provides further mechanistic support for the involvement of activin signaling in skeletal muscle, including evidence of SMAD3 phosphorylation and the expression of atrophy-related genes. The additional experiments examining the systemic effects of gingiva-derived activin A and the expanded analyses of muscle phenotypes also strengthen the overall conclusions.

The authors have also clarified several methodological issues, including the description of the datasets used for single-cell RNA-seq analysis and the experimental procedures for transcriptomic analyses. These revisions improve the transparency and reproducibility of the study.

Overall, the authors have responded appropriately to the reviewers' comments, and the manuscript has been significantly strengthened. In my opinion, the revised manuscript is suitable for publication in its current form.

- **Response:** We sincerely thank the reviewer for the positive feedback. We are pleased that our additional experiments and mechanistic evidence successfully addressed the concerns and strengthened the overall conclusions of the manuscript.

Reviewer #3:

- **Response:** We sincerely thank the reviewer for their time and valuable contribution to the review process, as well as for participating in this co-review initiative.

Reviewer #4:

The authors have done an exceptional job addressing all concerns raised in the previous round of review.

Key strengths of the revised manuscript:

1. Comprehensive experimental revisions: The authors have added functional assessments (grip strength, bone biomechanics), protein-level validation, and demonstrated activation of downstream Smad signaling in muscle tissue.

2. New high-quality scRNA-seq data: The replacement of public datasets with newly generated, high-resolution data from mice with confirmed muscle atrophy substantially strengthens the cellular localization findings.

3.Human validation: The addition of serum activin A measurements in periodontitis patients and epidemiological data linking periodontitis to reduced grip strength significantly enhances translational relevance.

4.Direct mechanistic evidence: The elegant FLAG-tagging experiment provides compelling direct evidence that gingiva-derived activin A reaches and signals in distant skeletal muscle, establishing the proposed periodontal-muscle crosstalk on solid experimental footing.

The authors have produced a rigorous, well-controlled, and conceptually novel study that establishes periodontitis as a previously unrecognized source of systemic activin A driving muscle atrophy.

- **Response:** We are deeply grateful for the reviewer's positive and encouraging evaluation. We especially appreciate the recognition of our efforts to provide direct mechanistic evidence for the periodontal-muscle crosstalk, which has significantly enhanced the impact of our study.